# Rabphilin-3A undergoes phase separation to regulate GluN2A mobility and surface clustering

Lei Yang[1,7], Mengping Wei[1,7], Yangzhen Wang[2,7], Jingtao Zhang[1], Sen Liu[1], Mengna Liu[3], Shanshan Wang[3], Ke Li[1], Zhaoqi Dong[1] & Chen Zhang ®[1,4,5,6] ⊠

N-methyl-D-aspartate receptors (NMDARs) are essential for excitatory neurotransmission and synaptic plasticity. GluN2A and GluN2B, two predominant Glu2N subunits of NMDARs in the hippocampus and the cortex, display distinct clustered distribution patterns and mobility at synaptic and extrasynaptic sites. However, how GluN2A clusters are specifically organized and stabilized remains poorly understood. Here, we found that the previously reported GluN2A-specific binding partner Rabphilin-3A (Rph3A) has the ability to undergo phase separation, which relies on arginine residues in its N-terminal domain. Rph3A phase separation promotes GluN2A clustering by binding GluN2A's C-terminal domain. A complex formed by Rph3A, GluN2A, and the scaffolding protein PSD95 promoted Rph3A phase separation. Disrupting Rph3A's phase separation suppressed the synaptic and extrasynaptic surface clustering, synaptic localization, stability, and synaptic response of GluN2A in hippocampal neurons. Together, our results reveal the critical role of Rph3A phase separation in determining the organization and stability of GluN2A in the neuronal surface.

N-methyl-D-aspartate receptors (NMDARs) are required for excitatory neurotransmission and the plasticity of excitatory synapses[1,2]. NMDARs function in the form of heterotetramers assembled from various combinations of the GluN1, GluN2A-D, and GluN3A-B subunits[3,4]. The NMDAR subunit combinations generally differ across brain regions, neuronal types, and developmental stages[5,6] and also determine the biophysical and pharmacological properties of NMDARs[7,8]. GluN2A and GluN2B are the predominant GluN2 NMDAR subunits in the cortical and hippocampal regions of the brain[5,9,10]. It is generally believed that synaptic NMDARs switch from a GluN2B-dominant type to a GluN2A-dominant type in response to neuronal activity and sensory experiences during development[11–13]. GluN2A- and GluN2B-containing NMDARs are thought to play distinct roles in

neuronal plasticity and in pathological conditions, such as Parkinson's disease, ischemia, and Huntington's disease[14–16]. Furthermore, the ratio of GluN2A/GluN2B subunits in NMDARs has been found to be altered in rat and primate models of Parkinson's disease and under levodopa therapy, which is associated with the development of dyskinesia[14,17–19]. Targeting GluN2A-containing NMDARs has been considered an approach for reducing dyskinesia under levodopa therapy[20,21].

The GluN2A and GluN2B subunits differ in terms of their subcellular localization and stability in the neuronal surface[22–24]. For instance, super-resolution microscopy has shown that NMDARs are organized as nanoscale clusters in the neuronal surface and that the nanoscale clusters of GluN2A and GluN2B are differentially distributed at both synaptic and extrasynaptic sites[25]. An electron microscopy

[1]School of Basic Medical Sciences, Beijing Key Laboratory of Neural Regeneration and Repair, Advanced Innovation Center for Human Brain Protection, Capital Medical University, Beijing 100069, China. [2]School of Life Sciences, Tsinghua University, Beijing 100084, China. [3]Peking-Tsinghua Center for Life Sciences, Academy for Advanced Interdisciplinary Studies, Peking University, Beijing 100871, China. [4]Chinese Institute for Brain Research, Beijing 102206, China. [5]State Key Laboratory of Translational Medicine and Innovative Drug Development, Nanjing 210000 Jiangsu, China. [6]Beijing Laboratory of Oral Health, Capital Medical University, Beijing 100050, China. [7]These authors contributed equally: Lei Yang, Mengping Wei, Yangzhen Wang. ⊠e-mail: czhang@188.com

study revealed that the density of GluN2A and GluN2B immunogold signals showed distinct distributions in the postsynaptic density[26]. Furthermore, NMDARs' membrane mobility is often determined by its subunit composition, as GluN2A is relatively immobile compared to GluN2B[27,28]. The C-terminal domain (CTD) of GluN2 subunits is believed to be the critical region for the subunit-specific regulation of GluN2 localization and trafficking[3]. Specifically, the association of membrane-associated guanylate kinases (MAGUKs) (such as PSD95) with the PDZ-binding motif of the GluN2B CTD determines the synaptic retention and nanoscale organization of GluN2B[25,29]. The phosphorylation of the PDZ-binding motif of GluN2B disrupts its association with MAGUKs and leads to the mobilization of GluN2B in response to neural activity[30]. Interfering with the MAGUK-GluN2A interaction disrupts the clustering and mobility of GluN2A, but leaves the nanoscale organization of GluN2A unchanged[25]. In addition, GluN2A remained synaptically localized when the PDZ-binding motif was disrupted[29]. Therefore, a PDZ-binding motif-independent mechanism may be responsible for the organization and stability of GluN2A, which remains poorly understood.

Recently, several studies have demonstrated that the phase separation of receptor-binding proteins participates in membrane receptor localization and clustering[31–33], providing key insights into the organization of receptors. The GluN2A-specific binding partner, Rabphilin-3A (Rph3A), first identified as a vesicle-associated protein that regulates SNARE-dependent vesicle release by interacting with SNAP25[34–36], has been revealed to specifically interact with the CTD (but not the PDZ-binding motif) of GluN2A[37]. Rph3A stabilizes GluN2A-containing NMDARs by forming a complex with GluN2A and PSD95. Disrupting the interactions between Rph3A and GluN2A and between Rph3A and PSD95 or knocking down Rph3A could suppress the surface expression and synaptic localization of GluN2A, the NMDAR-mediated current, and LTP induction, and negatively impact cognition[37,38]. The synaptic localization of Rph3A and its interaction with the GluN2A subunit were found to be increased in a rat model of Parkinson's disease and levodopa-induced dyskinesia[39].

In this work, we found that Rph3A was able to undergo protein phase separation to form liquid-like condensates. After phase separation, Rph3A condensed with the part of CTD (specifically covering amino acid (aa) 1244-1464 of the entire C-terminal tail of aa 839-1464; referred to as GluN2A-CTD from hereon) of the GluN2A subunit of the NMDAR. Furthermore, the ternary complex formed by Rph3A, GluN2A-CTD, and PSD95 promoted the phase separation of Rph3A. In hippocampal neurons, the phase separation of Rph3A decreased the mobility of GluN2A and maintained the surface clustering and synaptic localization of GluN2A in hippocampal neurons. Our findings provide a mechanism that explains the surface clustering and immobility of GluN2A.

## Results

### Rph3A forms condensates with liquid-like properties in the optoDroplet assay

To test the phase separation capability of the GluN2A-specific binding partners, we performed an analysis using the optoDroplet assay, which is used to assess the light-activated formation of protein droplets in live cells[40–42]. For these assays, we constructed clones expressing proteins of interest fused with the photoactivatable Cry2 protein and mCherry. After expressing these fusion proteins in HEK293 cells, we stimulated the live cells with blue light to induce self-association of the Cry2 protein, leading to an increase in the local concentration of the protein of interest. Proteins that are capable of phase separation are expected to form liquid-like droplets in cells upon light stimulation (Fig. 1a), whereas proteins that are incapable of phase separation will remain diffuse. FUS, an RNA-binding protein capable of phase separation[43], was selected as the positive control in the optoDroplet assay, and mCherry-Cry2 was used as the negative control.

Rph3A, BRAG2, NEDD4, and RNF10 were previously reported to bind specifically to GluN2A but not to GluN2B[37,44–46]. The analysis of the phase separation capability of previously reported GluN2A-specific binding proteins via the optoDroplet assay showed that the Rph3A fusion protein facilitated the formation of droplets in HEK293 cells, similar to the FUS protein[40], while the other GluN2A-specific binding proteins and the mCherry-Cry2 protein remained diffuse in cells upon photoactivation (Fig. 1b and Supplementary Fig. 1). Moreover, fluorescence recovery after photobleaching (FRAP) analysis revealed a rapid recovery ($t_{1/2} = 4.35$ s) of the fluorescence intensity in the Rph3A droplets (Fig. 1c). We also found that the droplets fused with each other within seconds after stimulation (Fig. 1d). These results suggest that the Rph3A fusion protein could form liquid-like droplets in cells.

To further assess whether intermolecular interactions between Rph3A molecules were involved in droplet formation, we extended the optoDroplet assay by coexpressing an Rph3A-EGFP fusion protein with an Rph3A-mCherry-Cry2 fusion protein in HEK293 cells, allowing the visualization of interactions between the different fusion proteins. We observed the formation of Rph3A-mCherry-Cry2 droplets upon photoactivation and identified the colocalization and condensation of Rph3A-EGFP and Rph3A-mCherry-Cry2 (Fig. 1e, left and 1f, left). However, when used as the negative control, EGFP was unable to colocalize or condense with the droplets formed by Rph3A-mCherry-Cry2 (Fig. 1e, middle and Fig. 1f, middle), and Rph3A-EGFP could not colocalize or condense with FUS-mCherry-Cry2 (Fig. 1e, right and 1f, right). These results reveal a possible intermolecular interaction between Rph3A molecules that promotes Rph3A-dependent droplet formation. The FRAP assay (Fig. 1g) and fusion events (Fig. 1h) further confirmed the liquid-like properties of Rph3A-EGFP droplets. Together, these results suggest that Rph3A proteins form droplets in the cells by liquid–liquid phase separation (LLPS).

### Functional domain mapping of Rph3A in phase separation

Since intrinsically disordered regions (IDRs) play a key role in phase separation[47,48], and the N-terminal domain (NTD) of Rph3A consists of two IDRs linked by a zinc finger (ZF) domain (Fig. 2a), it seemed likely that the observed LLPS properties of Rph3A were due to its NTD. To test this hypothesis, we generated several truncated Rph3A constructs (consisting of the NTD or CTD alone) and tested the ability of full-length (FL) Rph3A and truncated Rph3A constructs to form droplets in HEK293 cells using the optoDroplet assay. As expected, the NTD of Rph3A formed droplets upon light exposure, similar to FL Rph3A, whereas the distribution of the CTD of Rph3A remained diffuse in the cells (Fig. 2b). Furthermore, to determine which region in the NTD was responsible for the ability of Rph3A to form droplets, four additional constructs were generated (IDR1, IDR1 + ZF, IDR2, and ZF + IDR2), and their LLPS properties were assessed using an optoDroplet assay. Remarkably, the deletion of IDR2, the longer IDR, did not affect droplet formation, whereas the deletion of the shorter IDR1 abolished the ability of Rph3A to form droplets (Fig. 2c).

To further ascertain the key amino acids in IDR1 that determine the capability of Rph3A to undergo phase separation, we analyzed the sequences of IDR1 from different species and identified 9 conserved arginines (Arg) residues in the IDR1 sequence (Fig. 2d) that were previously reported to be involved in charge–charge interactions[32]. To investigate the role of these Arg residues in Rph3A-dependent phase separation, we mutated the Arg residues in IDR1 to alanine (Ala) residues (R9A) and assessed droplet formation using an optoDroplet assay. We found that IDR1-R9A failed to form droplets (Fig. 2e). Collectively, these results indicate that the Arg residues in the IDR1 of Rph3A might be required for Rph3A to undergo phase separation.

### Rph3A phase separation in vitro

To exclude the possibility that our results were compromised by a complicated cellular milieu in HEK293 cells, we purified EGFP-tagged

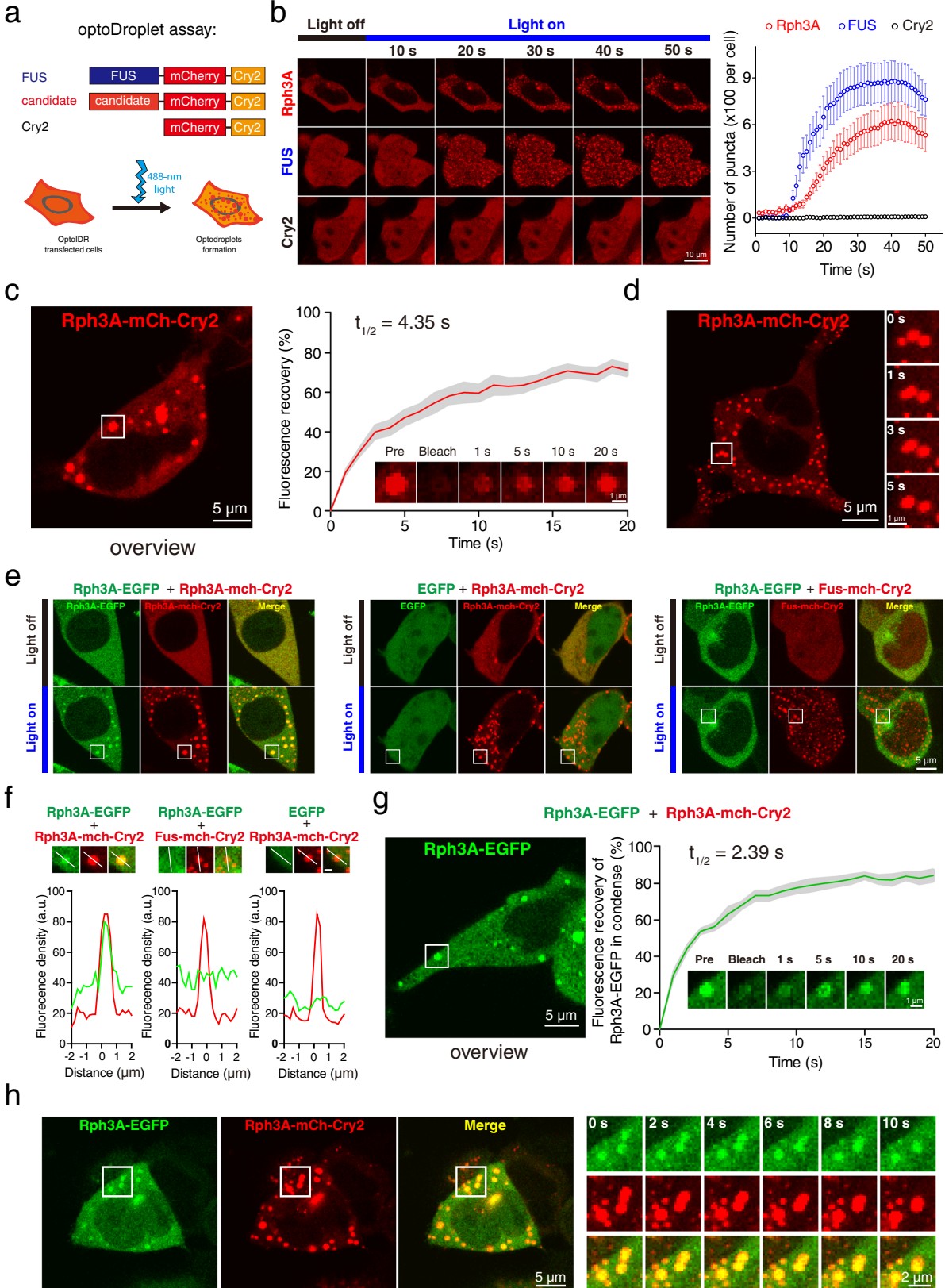

Rph3A and EGFP on its own from *E. coli* (Fig. 3f). The purified recombinant fusion proteins were added to a buffer with a physiological salt concentration containing 2% PEG8000, which mimics the crowded cellular environment[41,49]. At a concentration of 10 μM, Rph3A formed spherical GFP-positive droplets, as revealed by the differential interference contrast (DIC) and fluorescence microscopy, while EGFP at the

same concentration remained diffuse (Fig. 3a, middle and left). To test whether the Rph3A underwent phase separation without the presence of the crowding reagent, we concentrated the purified protein to 50 μM in a buffer containing a physiological salt concentration (150 mM NaCl) without a crowding reagent and we observed droplet formation (Fig. 3a, right). The size of the droplets formed by Rph3A

**Fig. 1 | Rph3A forms condensates with liquid properties in living cells.**
**a** Schematic illustration of the optoDroplet assay in HEK293 cells. Candidate genes were fused with mCherry and Cry2. The RNA-binding protein FUS was chosen as a positive control. Unfused mCherry-Cry2 was used as a negative control. Upon blue light exposure, proteins with phase-separation capacity formed condensates in cells. **b** OptoDroplet assay of Rph3A performed in HEK293 cells. The number of puncta per cell was counted and plotted. Data are displayed as the mean ± SEM (*n* = 6 cells per group). **c** Representative images and quantification of fluorescence recovery from the FRAP analysis of Rph3A droplets. The time point of bleaching was 0 s. The data are displayed as the mean ± SEM (*n* = 6 droplets). **d** Fusion events of droplets composed of Rph3A-mCherry-Cry2 in HEK293 cells captured by time-series imaging. **e** Rph3A-mCherry-Cry2 droplets condensed with Rph3A-EGFP after blue light stimulation. FUS and EGFP were substituted for Rph3A as negative controls. **f** Magnified images, in which the lines indicate the fluorescence intensity profiles of puncta in the white squares in e. Scale bar, 1 μm. **g** Representative images and quantification of fluorescence recovery from the FRAP analysis of Rph3A-EGFP droplets. The data are displayed as the mean ± SEM (*n* = 5 droplets). **h** Fusion events of Rph3A-EGFP and Rph3A-mCherry-Cry2 condensates in HEK293 cells stimulated with blue light. The image of cell at time point 0 s (left) and the magnified time-lapse images of the white squares in left (right) are showed. Source data and of **b**, **c** and **g** are provided in the Source Data file.

significantly decreased with a decreasing protein concentration (Fig. 3b) and an increasing salt concentration (Fig. 3c). Furthermore, the droplets tended to fuse together within 1 min when they wetted the surface of the glass coverslip (Fig. 3d). A FRAP assay further revealed that the Rph3A proteins in the droplets underwent an exchange with the molecules in the surrounding solution (Fig. 3e), suggesting that the droplets have liquid-like properties.

Additionally, we purified EGFP-tagged IDR1-only Rph3A and IDR1-R9A Rph3A from *E. coli* (Fig. 3f). The EGFP-tagged IDR1 of Rph3A exhibited the same capability to undergo phase separation as the FL Rph3A in vitro, but this effect was abolished by the R9A mutation. The sizes of the droplets formed by the different truncated Rph3A constructs decreased with a decreasing protein concentration (20 μM, 10 μM, and 5 μM) (Fig. 3g). These findings reveal that the Rph3A protein forms liquid-like droplets by itself in vitro and confirm that the previously identified Arg residues in IDR1 are essential for phase separation.

### Rph3A condenses with the CTD of GluN2A

Since Rph3A was previously reported to interact with GluN2A through the intracellular Glun2A CTD[37], we next investigated whether the CTD of GluN2A (GluN2A-CTD) could colocalize and condense with Rph3A in cells and in vitro. To this end, we coexpressed EGFP-tagged GluN2A-CTD (aa 1244–1464) with Rph3A-mCherry-Cry2 in HEK293 cells and induced droplet formation by photoactivation. GluN2A-CTD-EGFP showed a diffuse distribution before photoactivation and then colocalized and condensed with droplets composed of Rph3A-mCherry-Cry2 after photoactivation (Figs. 4a, b). The FRAP assay also revealed rapid fluorescence recovery of GluN2A-CTD-EGFP (Fig. 4c). The droplets containing GluN2A-CTD tended to fuse in seconds (Fig. 4d). These results suggest that the CTD of GluN2A could form liquid condensates aided by Rph3A in living cells.

To exclude the possibility that the complicated cellular milieu affected the cocondensation of GluN2A-CTD and Rph3A, we purified mCherry-tagged GluN2A-CTD and added the fusion protein to either the EGFP or the Rph3A-EGFP fusion protein. In the presence of EGFP, the distribution of GluN2A-CTD-mCherry remained diffuse (Fig. 4e, bottom), but GluN2A-CTD-mCherry formed droplets in the presence of Rph3A-EGFP (Fig. 4e, top). The distribution of the mCherry protein, used as a negative control, remained diffuse in the presence of Rph3A-EGFP (Fig. 4e, middle). A FRAP assay and an analysis of fusion events further confirmed the liquid-like properties of GluN2A-CTD droplets (Figs. 4f, g). These results demonstrate that the CTD of GluN2A could form liquid condensates aided by Rph3A in vitro.

To determine which of Rph3A's domains mediates its condensation with GluN2A, we tested the formation of droplets consisting of GluN2A-CTD with individual Rph3A deletion constructs (Fig. 4h). Similar to FL Rph3A, Rph3A-NTD was able to condense with GluN2A-CTD (Fig. 4h, left and 4i, left). The IDR1 + ZF construct, which lacked IDR2, still formed droplets upon photoactivation, but its ability to condense with GluN2A-CTD was abolished (Fig. 4h, right and 4i, right). Because the aa 1-179 of Rph3A, which contains the IDR1 + ZF sequence (aa 1-157), has been reported to interact with GluN2A-CTD[37], we further tested the condensation of GluN2A-CTD with aa 1–179 of Rph3A but observed no cocondensation (Supplementary Fig. 2a, b). A coimmunoprecipitation (Co-IP) assay revealed that the ablation of IDR2 significantly weakened the interaction of Rph3A with GluN2A-CTD, whereas IDR2-CTD, IDR1 + ZF and aa 1–179 of Rph3A were still bound to GluN2A-CTD (Supplementary Fig. 2c–e). Our results suggest that, in addition to aa 1–179 of Rph3A, IDR2 interacted with GluN2A-CTD, and the deletion of IDR2 impaired the interaction and condensation of Rph3A with GluN2A-CTD.

To further map the essential region of GluN2A-CTD that mediates its condensation with Rph3A, we tested the condensation of Rph3A with aa 1244–1348, aa 1244–1388, aa 1349–1388, aa 1349–1464, aa 1389–1464 and FL of GluN2A-CTD constructs, as illustrated in Supplementary Fig. 3a. Whereas FL GluN2A-CTD condensed with Rph3A, all the truncated constructs of GluN2A-CTD failed to condense with Rph3A (Supplementary Fig. 3b–g). Co-IP assays revealed that the truncated GluN2A-CTD constructs interacted more weakly with Rph3A than the FL Glun2A-CTD did (Supplementary Fig. 3h, i), which might have impaired the cocondensation. Thus, our results suggest that aa 1244–1464 of GluN2A-CTD are critical for the interaction and condensation of GluN2A with Rph3A.

### The GluN2A/PSD95 complex promotes the phase separation of Rph3A

Since Rph3A has been reported to interact with PSD95 to form a ternary complex with GluN2A and PSD95[37], we next sought to examine whether the interactions among Rph3A, GluN2A, and PSD95 regulate the phase separation of Rph3A. We expressed different combinations of EGFP-tagged Rph3A, mCherry-tagged GluN2A-CTD, and BFP-tagged PSD95 in HEK293 cells. The distribution of Rph3A, GluN2A-CTD, and PSD95 remained diffuse in cells when they were individually expressed (Fig. 5a). Surprisingly, GluN2A-CTD and PSD95 coexpression induced the formation of clusters in the cytoplasm, whereas the distribution of coexpressed GluN2A-CTD/Rph3A and PSD95/Rph3A remained diffuse. When Rph3A, GluN2A-CTD, and PSD95 were coexpressed in the cells, Rph3A was recruited to GluN2A-CTD/PSD95 clusters; the clusters formed by wild-type (WT) Rph3A with GluN2A-CTD/PSD95 were significantly larger than those formed by GluN2A/PSD95 as well as those formed by R9A Rph3A with GluN2A-CTD/PSD95 (Fig. 5b). A FRAP assay further revealed that the Rph3A condensates showed liquid-like properties (Supplementary Fig. 4). These results suggest that the GluN2A-CTD/PSD95 complex recruits Rph3A to further promote the phase separation of Rph3A, GluN2A-CTD, and PSD95.

To further confirm the regulation of Rph3A phase separation by the GluN2A-CTD/PSD95 complex in a cell-free system, we purified the recombinant EGFP-Rph3A, mCherry-GluN2A-CTD, and PSD95-BFP fusion proteins. Unlike the results obtained with the HEK293 cells, the mixture of GluN2A-CTD and PSD95 could not form condensates (Fig. 5c). The mixing of Rph3A with GluN2A-CTD or PSD95 induced the recruitment of GluN2A-CTD or PSD95 into Rph3A condensates; however, it did not increase the Rph3A droplet size (Figs. 5c, d,

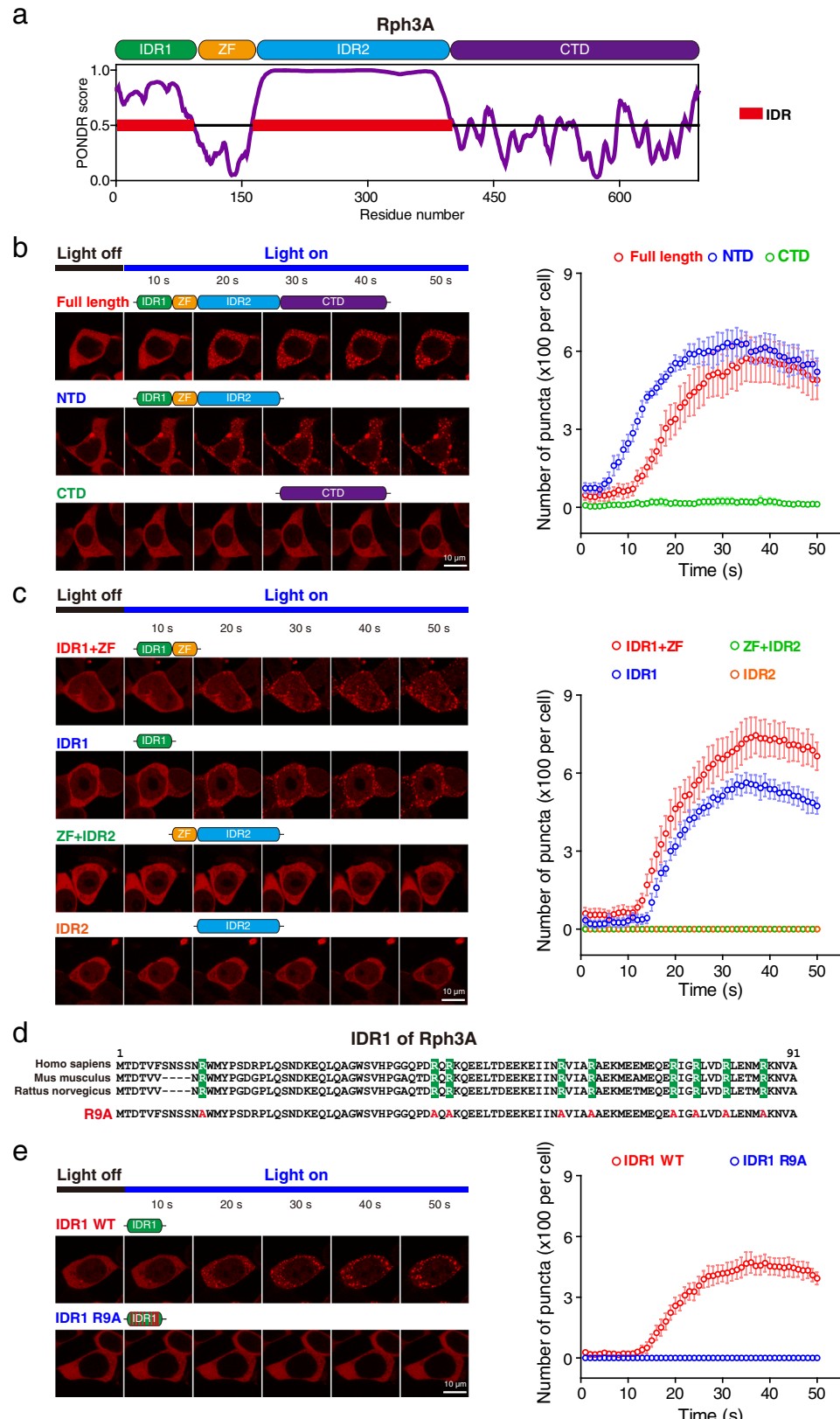

Supplementary Fig. 5a, b). When the three proteins were mixed together, the Rph3A droplet size was significantly increased compared with those composed of Rph3A alone (Figs. 5c, d and Supplementary Fig. 5c). Together, these findings reveal that the formation of a ternary complex consisting of Rph3A, GluN2A, and PSD95 promotes the phase separation of Rph3A.

**The phase separation of Rph3A reduces the mobility of GluN2A**

To further confirm Rph3A phase separation and determine its biological function in hippocampal neurons, we replaced endogenous Rph3A with either EGFP-tagged WT Rph3A or EGFP-tagged R9A mutant Rph3A in cultured hippocampal neurons by expressing mouse Rph3A shRNA and either EGFP-tagged WT Rph3A or R9A mutant Rph3A within

**Fig. 2 | Arg residues in IDR1 of Rph3A are essential for Rph3A phase separation.**
**a** Graphs showing the intrinsic disorder and domain arrangement of human Rph3A.
The y-axis shows the PONDER VSL2 score which indicates the extent of disorder,
and the x-axis is the amino acid position. Intrinsically disordered regions (IDRs) are
marked in red. The Rph3A amino acid sequence is divided into four parts:
IDR1, the zinc finger (ZF) domain, IDR2, and the C-terminal domain (CTD).
**b**, **c** Representative images and quantification of condensates formed by

truncated Rph3A in the optoDroplet assay. The data are displayed as the
mean ± SEM ($n = 6$ cells per group). **d** Alignment of IDR1 sequences from
different species. The conserved Arg residues, marked in green, were muta-
ted to Ala residues (R9A). **e** Representative images and quantification of
condensates formed by WT Rph3A and R9A Rph3A in HEK293 cells. The data
are displayed as the mean ± SEM ($n = 6$ cells per group). Source data of
**b**, **c** and **e** are provided in the Source Data file.

a single plasmid. EGFP-tagged WT Rph3A formed puncta, whereas the
distribution of R9A mutant Rph3A tended to remain diffuse in neu-
ronal dendrites (Fig. 6a). To quantify the synaptic localization of
Rph3A puncta, we used a Homer1 antibody to label the excitatory
synapses. We found that nearly 50% of the Rph3A WT puncta were
synaptic, which were colocalized with Homer1 (Fig. 6a). The synaptic
puncta were significantly larger than the extrasynaptic puncta (Fig. 6a).
The liquid-like properties of the puncta were further verified by a FRAP
assay (Fig. 6b). These results suggest that Rph3A underwent phase
separation in neuronal dendrites and that synaptic localization pro-
moted the phase separation of Rph3A.

Previous research has shown that Rph3A clusters with GluN2A at
dendrites and spines[37] and that GluN2A is less mobile than GluN2B. As
such, our next step was to investigate whether the phase separation of
Rph3A regulates the mobility of GluN2A clusters in hippocampal
neurons. To this end, we performed a FRAP assay using recombinant
GluN2A tagged with pH-sensitive super-ecliptic pHluorin (SEP), which
fluoresces brightly at the cell surface but not in the acidic intracellular
environment[50,51], in endogenous Rph3A-knockdown and WT/R9A
Rph3A-expressing hippocampal neurons (Figs. 6c and 6d). The recovery
rate was faster in the Rph3A KD group ($t_{1/2}$ at 2.231 ± 0.2916 min) than in
the scramble group ($t_{1/2}$ at 4.804 ± 1.054 min), and this change was
rescued by WT Rph3A but not by R9A Rph3A (Fig. 6e). Additionally, the
proportion of recovered SEP-GluN2A fluorescence increased from
16.34 ± 2.706% in the scramble group to 45.50 ± 2.160% in the Rph3A KD
group, and this change was rescued by WT Rph3A, but not by R9A
Rph3A (Fig. 6f). Unlike the results obtained for GluN2A, the mobility of
SEP-GluN2B was not changed by Rph3A knockdown (Supplementary
Fig. 6a–d). Thus, our observations strongly indicate that the phase
separation of Rph3A specifically reduces the mobility of GluN2A at
dendrites in hippocampal neurons.

To confirm the effect of Rph3A phase separation on the mobility
of GluN2A, we performed a FRAP assay of EGFP-tagged GluN2A-CTD in
HEK293 cells expressing BFP-tagged PSD95, EGFP-tagged GluN2A-
CTD, and mCherry-tagged Rph3A (Supplementary Fig. 7a–d). Com-
pared with the mCherry group and the Rph3A R9A group, the EGFP
fluorescence recovery rate was significantly slower in the Rph3A WT
group (Supplementary Fig. 7e). The proportion of EGFP fluorescence
recovery was also lower in the Rph3A WT group (Supplementary
Fig. 7f). These results indicate that Rph3A with the capacity for phase
separation, could reduce the mobility of GluN2A in the HEK293 cell,
which supports our findings in hippocampal neurons.

**The phase separation of Rph3A maintains both the synaptic and
extrasynaptic surface clustering of GluN2A and the synaptic
localization of GluN2A**
Because Rph3A has been reported to maintain the surface expression
of GluN2A in hippocampal neurons[37], we next sought to investigate
whether the capacity of Rph3A to undergo phase separation is
responsible for its ability to maintain GluN2A clustering on the mem-
brane. For this purpose, we quantified the fluorescence intensity and
density of synaptic and extrasynaptic GluN2A clusters in Rph3A
knockdown neurons and neurons in which WT Rph3A or R9A Rph3A
expression was rescued (Fig. 6g). We found that the knockdown of
Rph3A decreased the density of both synaptic and extrasynaptic sur-
face GluN2A clusters, and the fluorescence intensity of synaptic
GluN2A clusters decreased in the Rph3A knockdown group compared

with the control group. WT Rph3A rescued these phenotypes, while
the R9A mutant Rph3A could not (Figs. 6h, i). The percentage of
synaptic GluN2A decreased in the Rph3A knockdown group, and the
synaptic localization of GluN2A in the WT Rph3A-reexpressing group
was stronger than that in the R9A Rph3A-reexpressing group (Fig. 6j).
Unlike the findings obtained for GluN2A clusters, the fluorescence
intensity and density of the synaptic and extrasynaptic GluN2B clusters
were not influenced by Rph3A knockdown (Supplementary Fig. 6e, f).
These results revealed that the phase separation of Rph3A maintained
both the synaptic and extrasynaptic surface GluN2A clustering and
promoted the synaptic localization of GluN2A.

Since the GluN2A/PSD95 complex has been reported to deter-
mine the synaptic localization of GluN2A[52,53], we next examined
whether the phase separation of Rph3A could modify the formation
of the GluN2A/PSD95 complex in the neuronal surface. Hence, we
quantified the fluorescence intensity and density of surface GluN2A
clusters that did or did not colocalize with PSD95 and the percentage
of GluN2A that colocalized with PSD95 in Rph3A knockdown neu-
rons, and neurons in which WT Rph3A or R9A Rph3A expression was
rescued (Supplementary Fig. 8a). We found that knocking down
Rph3A significantly decreased the intensity and density of the
GluN2A that colocalized with PSD95, as well as the density of the
GluN2A that did not colocalize with PSD95, when compared with
the control group. WT Rph3A rescued these phenotypes, whereas
R9A Rph3A could not (Supplementary Fig. 8c–f). The percentage of
GluN2A that colocalized with PSD95 was decreased in the Rph3A
knockdown group, and the colocalization of GluN2A with PSD95 in
the WT Rph3A-reexpressing group was stronger than that in the R9A
Rph3A-reexpressing group (Supplementary Fig. 8b). As Rph3A was
reported to regulate PSD95 clustering[37], we further quantified the
intensity and density of PSD95 clusters, where we found that the
density of the PSD95 clusters in the WT Rph3A-reexpressing group
was higher than that in the R9A Rph3A-reexpressing group (Sup-
plementary Fig. 8g, h). Thus, our findings indicate that phase
separation of Rph3A could maintain the GluN2A/PSD95 complex and
PSD95 clustering in the neuronal surface.

To further clarify the effect of Rph3A phase separation on the
interaction between GluN2A and PSD95, we performed a Co-IP assay of
GluN2A-CTD with PSD95 in the presence of either WT or the R9A
mutant Rph3A. The Co-IP assay revealed that the R9A mutant did not
completely abolish but instead significantly decreased the promotion
of GluN2A/PSD95 complex formation by WT Rph3A (Supplementary
Fig. 8i), whereas the R9A mutant did not influence the interaction
between Rph3A and PSD95 or between Rph3A and GluN2A (Supple-
mentary Fig. 8j and 2e). Thus, our observations suggested that the
phase separation of Rph3A plays a critical role in the interaction
between GluN2A-CTD and PSD95, the formation of the GluN2A/PSD95
complex and the clustering of PSD95 in neuronal dendrites, which
might be responsible for the synaptic localization of GluN2A.

Further examination of whether the phase separation of Rph3A
modulated spine density was performed, as the Rph3A/GluN2A/PSD95
complex has been shown to regulate dendritic spine numbers[37]. We
monitored dendritic spine morphology by expressing mCherry in
Rph3A knockdown neurons and neurons in which WT Rph3A or R9A
Rph3A expression was rescued, where we observed a loss of spines in
the Rph3A knockdown group (Supplementary Fig. 9). The R9A mutant
Rph3A did not rescue dendritic spine loss, while WT Rph3A did,

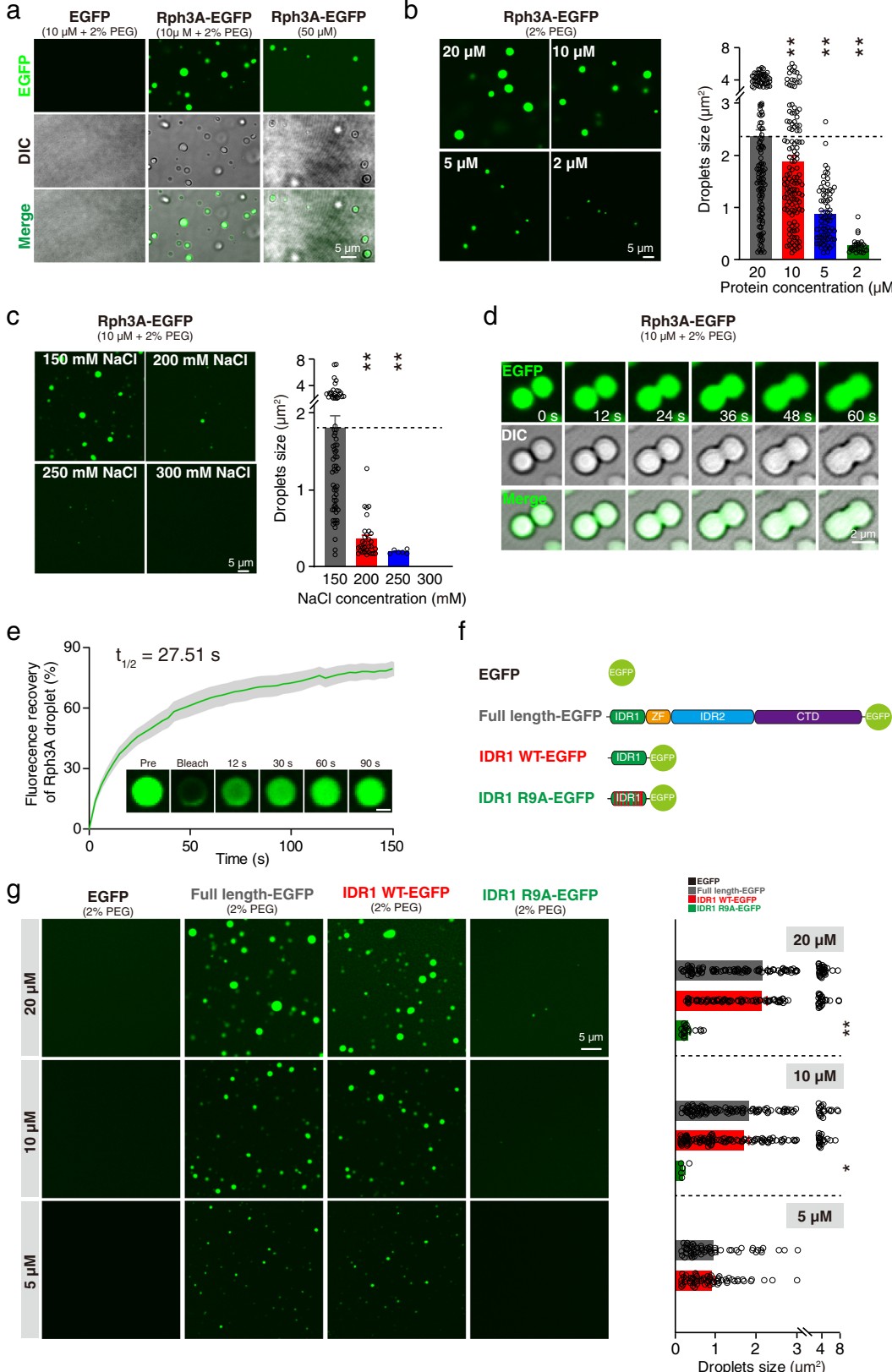

indicating the association of Rph3A phase separation with normal spine density in cultured hippocampal neurons.

We next examined the functional consequences of disrupting Rph3A phase separation on GluN2A-dependent synaptic transmission. GluN2A-dependent action-potential evoked excitatory postsynaptic currents (eEPSCs) were recorded for cultured hippocampal neurons

treated with AMPAR, GABAR, and GluN2B blockers. Rph3A knockdown significantly decreased the amplitude of GluN2A-dependent eEPSCs from $108.8 \pm 11.54$ pA to $53.18 \pm 8.476$ pA. WT Rph3A rescued this phenotype, while R9A Rph3A did not (Figs. 6k, l). The paired pulse ratio of GluN2A-dependent eEPSCs did not differ among the groups (Supplementary Fig. 10), indicating no change in presynaptic neurotransmitter

**Fig. 3 | Rph3A undergoes phase separation in a cell-free system. a** The images of EGFP alone (left), the full-length Rph3A fusion protein at a concentration of 10 μM in buffer containing 2% PEG8000 (middle) or a concentration of 50 μM without the crowding reagent (right). **b** Representative images and size data for droplets composed of the Rph3A fusion protein at different concentrations. The data are displayed as the mean ± SEM (20 μM: $n = 145$ droplets; 10 μM: $n = 125$ droplets; 5 μM: $n = 72$ droplets; 2 μM: $n = 28$ droplets, $^{**}p < 0.01$, one-way ANOVA followed by Tukey's multiple comparisons test). **c** Representative images and size data for droplets composed of the Rph3A fusion protein at 10 μM in buffer with different salt concentrations. The data are displayed as the mean ± SEM (150 mM: $n = 72$ droplets; 200 mM: $n = 31$ droplets; 250 mM: $n = 6$ droplets, 300 mM: no droplets, $^{**}p < 0.01$, one-way ANOVA followed by Tukey's multiple comparisons test). **d** Fusion events of droplets formed by the Rph3A fusion protein. Time-lapse images of fusing droplets are showed. **e** Representative images and quantification of fluorescence recovery from the FRAP analysis of droplets composed of the Rph3A fusion protein. The data are displayed as the mean ± SEM ($n = 7$ droplets). Scale bar, 1 μm. **f** Schematic illustration of recombinant Rph3A-EGFP fusion proteins. **g** Representative images and size data for droplets composed of full-length and truncated Rph3A fusion proteins at different protein concentrations. The data are displayed as the mean ± SEM (20 μM: $n = 122$ droplets (FL); $n = 101$ droplets (IDR1); $n = 17$ droplets (ID1R9A), 10 μM: $n = 114$ droplets (FL); $n = 120$ droplets (IDR1); $n = 5$ droplets (ID1R9A), 5 μM: $n = 54$ droplets (FL); $n = 60$ droplets (IDR1); no droplets (ID1R9A), no droplets in EGFP group, $^{*}p < 0.05$, $^{**}p < 0.01$, one-way ANOVA followed by Tukey's multiple comparisons test). Source data and $p$ values of **b**, **c**, **e** and **g** are provided in the Source Data file.

release. These data suggest that the phase separation of Rph3A could maintain normal GluN2A-dependent synaptic transmission. Altogether, these results indicate that disruption of the phase separation of Rph3A impairs the surface clustering and function of GluN2A.

## Restoring phase separation of mutant Rph3A reinstates its effect on surface clustering and mobility of GluN2A

To validate that the phase separation of Rph3A promotes the surface clustering and stability of GluN2A, we restored the phase separation of Rph3A R9A by fusing the disordered N-terminus of DDX4 (aa 1-236) (dND), which was reported to drive liquid-liquid phase separation[54], to the N-terminus of R9A Rph3A (IDR1 only or FL) (Figs. 7a, c). The OptoDroplet assay showed that dND fused R9A-IDR1 formed droplets after stimulation in HEK293 cells (Fig. 7b), and dND fused EGFP-tagged R9A mutant Rph3A formed puncta at both synaptic and extrasynaptic sites in neuronal dendrites (Fig. 7c), suggesting that dND fusion restores the phase separation of Rph3A R9A.

Next, we tested the effect of dND fused Rph3A R9A on the mobility and surface clustering of GluN2A. We performed a FRAP assay of SEP-GluN2A in the endogenous Rph3A-knockdown and Rph3A WT, Rph3A R9A, dND fused Rph3A R9A, or dND-expressing hippocampal neurons (Figs. 7d, e). We found that the recovery rate and mobile fraction of SEP-GluN2A in the dND fused Rph3A R9A group were comparable to those in the Rph3A WT group, while those in the Rph3A R9A and dND groups were not (Figs. 7f, g). We assessed the surface clustering of GluN2A and found that the density of synaptic and extrasynaptic GluN2A clusters and intensity of extrasynaptic GluN2A clusters, as well as the synaptic localization of GluN2A in the dND fused Rph3A R9A group, were comparable to those in the Rph3A WT group, while those in the Rph3A R9A and dND groups were not (Fig. 7h–k). Overall, our data demonstrates that the effect of Rph3A on the surface clustering and mobility of GluN2A depends on the phase separation of Rph3A.

## Discussion

In this study, we found that the GluN2A-specific binding partner Rph3A undergoes phase separation in HEK293 cells, cell-free systems, and neurons. Arginine residues in the N-terminus of Rph3A are essential for its phase separation capability. The phase separation of Rph3A can recruit GluN2A-CTD, PSD95, and the GluN2A-CTD/PSD95 complex, while the GluN2A/PSD95 complex in turn promotes Rph3A phase separation. Moreover, we found that the phase separation of Rph3A stabilizes GluN2A-NMDARs at synaptic and extrasynaptic sites in the neuronal surface and promotes the synaptic localization of GluN2A, revealing a mechanism of NMDAR organization and functional regulation.

Rph3A was first identified as an effector of Rab3a, a small GTP-binding protein enriched in synaptic vesicles that regulates synaptic vesicle release at presynaptic sites[55]. Rph3A interacts with Rab3a via a Rab-binding domain (RBD) (aa 40-170) in its N-terminus[56,57]. Here, we showed that IDR1 (aa 1–91), which overlaps with part of the RBD, is critical for the self-association of Rph3A. Since IDR1 lacks the zinc finger and SGAWFF motifs that are essential for Rab3a binding[58], the self-association of Rph3A might not depend on Rab3a binding. A study of the crystal structure of the RBD–Rab3a complex indicated that the binding of Rab3a could prevent the aggregation of the RBD and that the RBD alone appeared to be flexible and tended to self-associate[58]. Moreover, IDR1 is enriched with charged residues; there are 14 conserved alkaline residues (9 Arg residues and 5 Lys residues) and 17 conserved acidic residues (11 Glu residues and 6 Asp residues) in IDR1. These charged residues likely form charge–charge interactions between different Rph3A molecules, which should promote the self-association of Rph3A. Furthermore, a crystal structure study also revealed that several Arg and Glu residues of IDR1 are in close contact with Rab3a in the RBD-Rab3a complex[58]. Our data shows that Rab3a prevents the phase separation of Rph3A and impairs the interaction between GluN2A and Rph3A (Supplementary Fig. 11). The physiological function of the Rab3a-mediated suppression of Rph3A phase separation needs to be elucidated in future research. Rph3A was previously found to bind SNAP-25, a member of the SNARE complex, via a C2B domain in the C-terminus of Rph3A[35,36]. Rph3A was also found to regulate the SNARE-dependent repriming of synaptic vesicles by interacting with SNAP-25[35]. Whether Rph3A phase separation is involved in its presynaptic function needs to be carefully examined.

Rph3A has been reported to form a ternary complex with GluN2A and PSD95 to stabilize GluN2A at synaptic sites[37]. Our findings provide a mechanism for ternary complex formation and GluN2A stabilization: the self-association of Rph3A promotes ternary complex formation, and the ternary complex promotes the self-association of Rph3A, which in turn makes the ternary complex more stable. Considering that Rph3A interacts with the CTD of GluN2A and the PDZ3 domain of PSD95[37] and that GluN2A interacts with the PDZ1 and PDZ2 domains of PSD95[59], the multivalent interaction between Rph3A, GluN2A, and PSD95 might promote Rph3A phase separation. Our results show that PSD95 and GluN2A-CTD formed clusters in HEK293 cells but were diffuse in a cell-free system (Fig. 5), suggesting that the GluN2A/PSD95 complex might also undergo phase separation with the help of unknown factors in HEK293 cells, which is worth examining in the future.

Postsynaptic scaffolding proteins, such as PSD-95, GKAP, Shank, and Homer, form the postsynaptic density by undergoing phase separation with liquid-like properties, providing a platform for the clustering of membrane proteins, such as ion channels, adhesion molecules, and neurotransmitter receptors[60]. As the clustered membrane proteins are not homogeneous but are enriched in nanodomains in the postsynaptic density, investigating how membrane proteins are distinctly organized on a given scaffold platform may yield useful insights. For instance, PSD-95 reportedly forms a liquid condensate with TARP γ−2 to assemble AMPA receptors (AMPARs) in the postsynaptic density[32]. This is believed to be the mechanism responsible for AMPAR nanodomain formation. NMDARs are also organized in the neuronal surface in a nano-scaled manner[25]. Our study suggests that Rph3A might

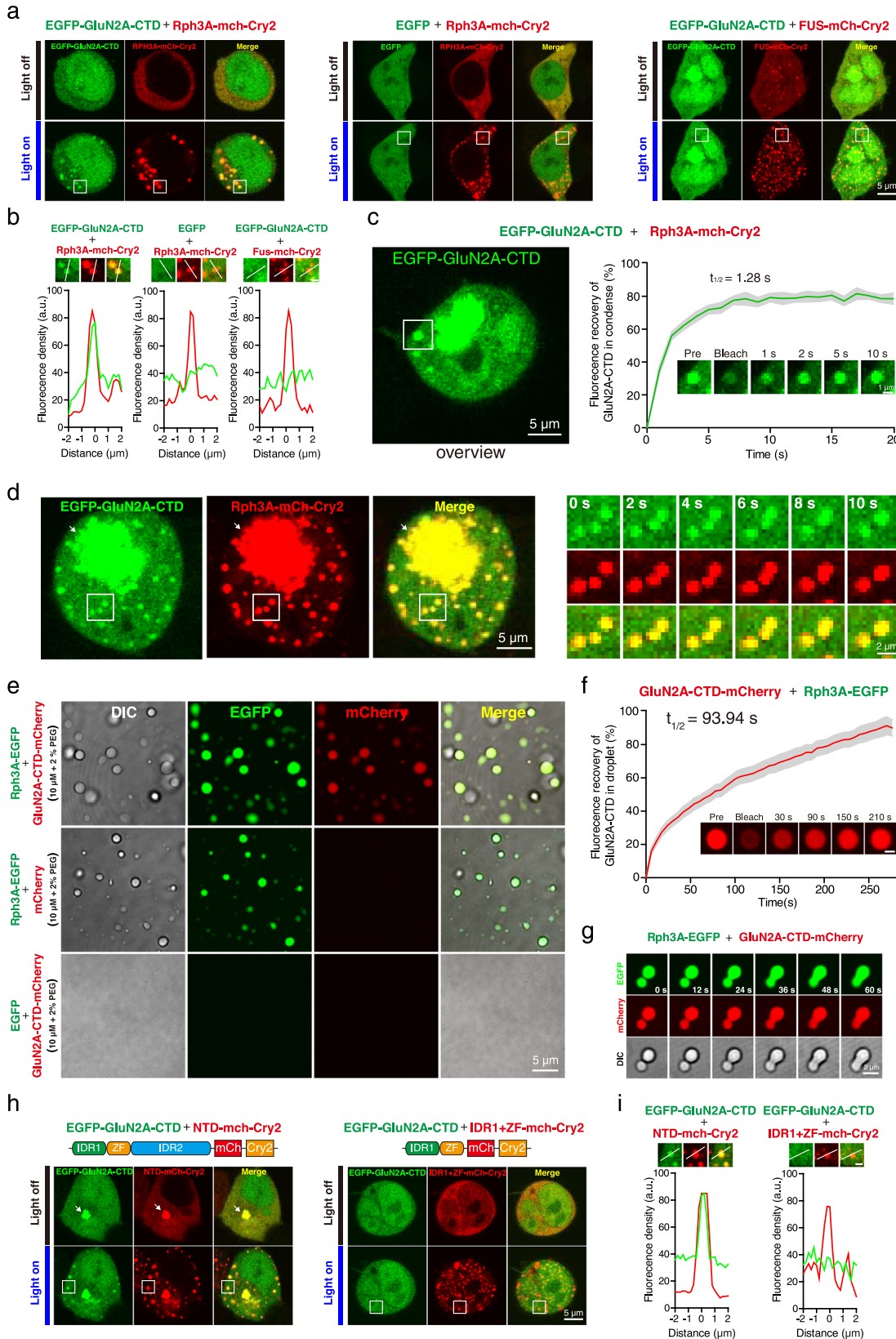

play a key role in this process: Rph3A condensation enriches GluN2A in the postsynaptic density formed by the phase separation of PSD95 with other scaffold proteins, which might be the mechanism responsible for the nanoscale organization of GluN2A-NMDARs at postsynaptic sites. As previously shown, GluN2A-containing NMDARs can also undergo clustering without colocalization with PSD95 clusters at extrasynaptic sites[25,61,62], and Rph3A localizes to dendritic spines and shafts[37]. We also observed the extrasynaptic phase separation of Rph3A in this study (Fig. 6a), and the disruption of Rph3A phase separation also induced the remodeling of extrasynaptic GluN2A clusters. The phase separation of Rph3A might explain how extrasynaptic GluN2A is organized in the neuronal surface.

**Fig. 4 | Rph3A condenses with the CTD of GluN2A. a** Rph3A droplets condensed with the CTD of GluN2A in the optoDroplet assay. **b** Magnified images, in which the lines indicate the fluorescence intensity profiles of condensates in the white squares in a. Scale bar, 1 μm. **c** Representative images and quantification of fluorescence recovery from the FRAP analysis of GluN2A-CTD condensed with Rph3A droplets. The data are displayed as the mean ± SEM (*n* = 6 puncta). **d** Fusion events of GluN2A-CTD condensed with Rph3A droplets in HEK293 cells stimulated with blue light. The image of cell at time point 0 s (left) and the magnified time-lapse images of the white squares in left (right) are showed. The large punctum (white arrow) is occasional preactivation. See also Supplementary Fig. 12. **e** The recombinant GluN2A-CTD mCherry fusion protein formed droplets with Rph3A-EGFP at a concentration of 10 μM in buffer containing 2% PEG8000 (top) but showed a diffuse distribution in buffer without Rph3A (bottom). The mCherry protein diffused in the presence of Rph3A-EGFP at the same concentration (middle). **f** Representative images and quantification of fluorescence recovery from the FRAP analysis of droplets composed of GluN2A-CTD in the presence of Rph3A. The data are displayed as the mean ± SEM (*n* = 8 droplets). Scale bar, 1 μm. **g** Fusion events of droplets composed of GluN2A-CTD in the presence of Rph3A. Time-lapse images of fusing droplets are showed. **h** Droplets of Rph3A containing IDR2 condensed with GluN2A-CTD. The big punctum (white arrow) is occasional preactivation foci. See also Supplementary Fig. 12. **i** Magnified images, in which the lines indicate the fluorescence intensity profiles of condensates in the white squares in h. Scale bar, 1 μm. Source data of c and f are provided in the Source Data file.

In this study, we found that disrupting the phase separation of Rph3A impaired the stabilization of synaptic and extrasynaptic GluN2A. Rph3A has been reported to stabilize synaptic GluN2A by blocking endocytosis[37]. These results suggest that disrupting the phase separation of Rph3A might result in the endocytosis of GluN2A in the neuronal surface. The phase separation of certain proteins could form a condense that excludes certain proteins; for instance, the condense formed by the phase separation of excitatory postsynaptic scaffolds could exclude the inhibitory postsynaptic scaffolds Gephyrin[33]. We posit that the phase separation of Rph3A might exclude the proteins that drive the endocytosis of GluN2A, which could then block the endocytosis of GluN2A and stabilize the GluN2A in the neuronal surface.

Considering that the GluN2 subunit composition of NMDARs is altered during development and under disease conditions[17–19], the process of Rph3A phase separation could be dynamic and might be modified along with changes in the GluN2 subunit composition. Posttranslational modifications, such as phosphorylation, have been reported to regulate phase separation. For instance, synapsin, which captures lipid vesicles and forms vesicle clusters via phase separation, can be phosphorylated by CaMKII, a serine/threonine-specific protein kinase in the CNS, resulting in the dispersal of synapsin and vesicle clusters[49]. CaMKII can also phosphorylate SAPAP, a component of PSD, and enhance PSD condensation[63]. Rph3A is another target of CaMKII and PKA[64,65]. However, the phosphorylation site is located in the IDR2 region[64], which is not essential for the phase separation of Rph3A. Hence, it would be interesting to examine whether the phase separation of Rph3A might be modified by phosphorylation or other posttranslational modifications and whether such posttranslational modifications are involved in the regulation of the GluN2 subunit composition during development and disease pathogenesis.

## Methods

### Cloning

The coding sequences of Rph3A, BRAG2, Nedd4, and RNF10 were amplified from a hORFeome V8.1 library by PCR using PrimSTAR Max DNA Polymerase (Takara, R045B). To generate optoDroplet plasmids, the coding sequences of selected proteins, mCherry, and Cry2 were fused and cloned into the pcDNA 3.1 vector using the pEASY-Uni Seamless Cloning and Assembly Kit (TransGen Biotech, CU101-01) according to the manufacturer's instructions.

The coding sequences of GluN2A, GluN2B, and PSD95 were amplified from cDNA from the C57BL mouse hippocampus. All amplified sequences were verified using DNAMAN (Version 9) to be identical to sequences published in the NCBI database (GluN2A: NM_008170.2 [Mus musculus glutamate receptor, ionotropic, NMDA2A (epsilon 1) (Grin2 - Nucleotide - NCBI (nih.gov)], GluN2B: NM_008171.3 [Mus musculus glutamate receptor, ionotropic, NMDA2B (epsilon 2) (Grin2 - Nucleotide - NCBI (nih.gov)], PSD95: NM_001109752.1 [Mus musculus discs large MAGUK scaffold protein 4 (Dlg4), transcript v - Nucleotide - NCBI (nih.gov)]. The sequences of SEP (Addgene, #24002), mCherry, EGFP, BFP, and Cry2 (Addgene, #101221) were amplified from preexisting plasmids. To generate eukaryotic expression vectors, the sequences of Rph3A, GluN2A, GluN2B, EGFP, and mCherry were cloned into the FUGW3 vector using seamless cloning. To generate prokaryotic expression vectors, the sequences of Rph3A, GluN2A, PSD95, EGFP, BFP and mCherry were fused and cloned between the BamHI and XhoI sites in frame with a His tag in the pET28a vector.

The shRNA plasmid for mouse Rph3A (Y12285) was purchased from OBiO Technology (Shanghai) Corp., Ltd. The target site was 5′-GCGCTTGAAACATTGGTAT-3′. The Rph3A WT and Rph3A R9A sequences were cloned into shRNA plasmids to obtain Rph3A-re-expression plasmids.

### Cell culture and transfection

HEK293 cells were obtained from ATCC (CRL-1573) and authenticated by STR profiling by the supplier. HEK293 cells were cultured in a 37 °C incubator supplied with 5% CO2. Transfection of the DNA was performed using polyethylenimine (PEI) (Polysciences: 24765). Brifely, the DNA and PEI were mixed and add to Opti-MEM buffer (Gibco: 31985070). After incubation at room temperature for 30 min, the mixture was added to the cultured HEK293 cells. The cells were grown for 2 days before experiments were performed.

Euthanasia of P0 pups were performed using decapitation after anesthetization on ice for 2 min. Hippocampal tissues from P0 pups were dissected and digested with 0.25% trypsin (Gibco, 25200072) at 37 °C for 12 min. Neurons were plated on glass coverslips precoated with poly-D-lysine and cultured at 37 °C in 5% CO2 for 14 days before the experiments. The cultured neurons were transfected with plasmids at 10 days in vitro (DIV 10) with calcium phosphate. Briefly, DNA and Ca$^{2+}$ were mixed and added to HBS buffer. After incubation for 30 min at room temperature, the mixture was added to the cultured neurons, and incubation was conducted at 37 °C for another 30 min. After two washes with culture medium, the neurons were grown in an incubator for 4 days before experiments were performed.

### OptoDroplet assay

A total of 300,000 HEK293 cells were plated on glass coverslips in 35-mm dishes 1 day before transfection. The HEK293 cells were then transfected with optoDroplet plasmids using PEI transfection reagent. After 2 days of transfection, light activation and imaging were performed using an Olympus FV3000 microscope with a 60X objective. Droplet formation was induced with 488-nm laser pulses applied at 1 s intervals during imaging. mCherry fluorescence was also captured every 1 s. The number of droplets in a cell was quantified using ImageJ (1.53k). In the cocondensation optoDroplet assay, the optoDroplet plasmids were cotransfected with EGFP-fused plasmids. EGFP and mCherry fluorescence were captured every 1 s during imaging.

### Protein purification

cDNAs encoding Rph3A, GluN2A-CTD (aa 1244–1464), and PSD95 were fused with the EGFP/mCherry/BFP coding sequence and cloned into the pET28a expression vector. A His-tag was fused to the

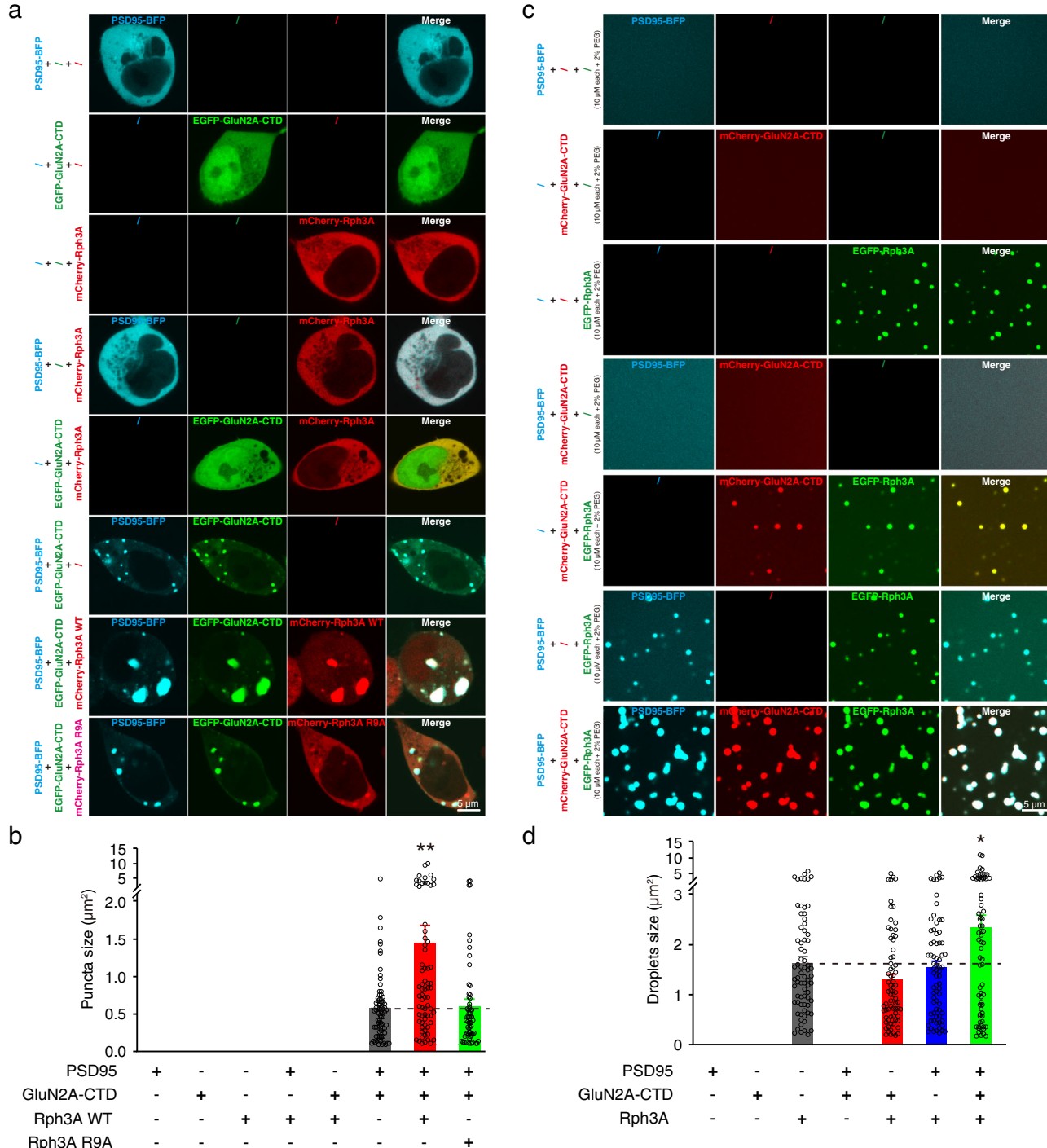

**Fig. 5 | The GluN2A/PSD95 complex promotes the phase separation of Rph3A.**
**a** Representative images of HEK293 cells transfected with different combinations of PSD95, GluN2A-CTD, WT, and R9A Rph3A. **b** Sizes of puncta composed of different combinations of the proteins in A. The data are displayed as the mean ± SEM (PSD95/GluN2A-CTD: $n = 78$ puncta; PSD95/GluN2A-CTD/Rph3A WT: $n = 73$ puncta; PSD95/GluN2A-CTD/Rph3A R9A: $n = 66$ puncta, $^{**}p < 0.01$, one-way ANOVA followed by Tukey's multiple comparisons test). **c** Representative images of droplets formed by different combinations of PSD95, GluN2A-CTD, WT, and R9A Rph3A recombinant fusion proteins in a cell-free system. **d** Sizes of droplets composed of different combinations of proteins in **c**. The data are displayed as the mean ± SEM ($n = 73$ droplets for each group, $^{*}p < 0.05$, one-way ANOVA followed by Tukey's multiple comparisons test). Source data and $p$ values of **b** and **d** are provided in the Source Data file.

C-terminus of the protein of interest. All constructs were sequenced to verify sequence identity. The fusion proteins were expressed in BL21 (DE3) cells. Specifically, cells transformed with pET28a were grown in medium containing kanamycin at 37 °C. After 16 h, the cells were split 1:30 in 300 ml of fresh medium and grown to the proper density (OD = 0.6). The expression of the protein of interest

was then induced in the cells by the addition of 50 mM IPTG. The cells were grown overnight at 18 °C and collected by centrifugation at $6000 \times g$ for 20 min at 4 °C. The pellets were frozen and stored at −80 °C or resuspended in 15 ml of Buffer A (50 mM Tris-HCl (pH 7.6), 500 mM NaCl, 1 mM EDTA, and 5% glycerol) containing 1 mM PMSF and protease inhibitors (Roche, 11873580001).

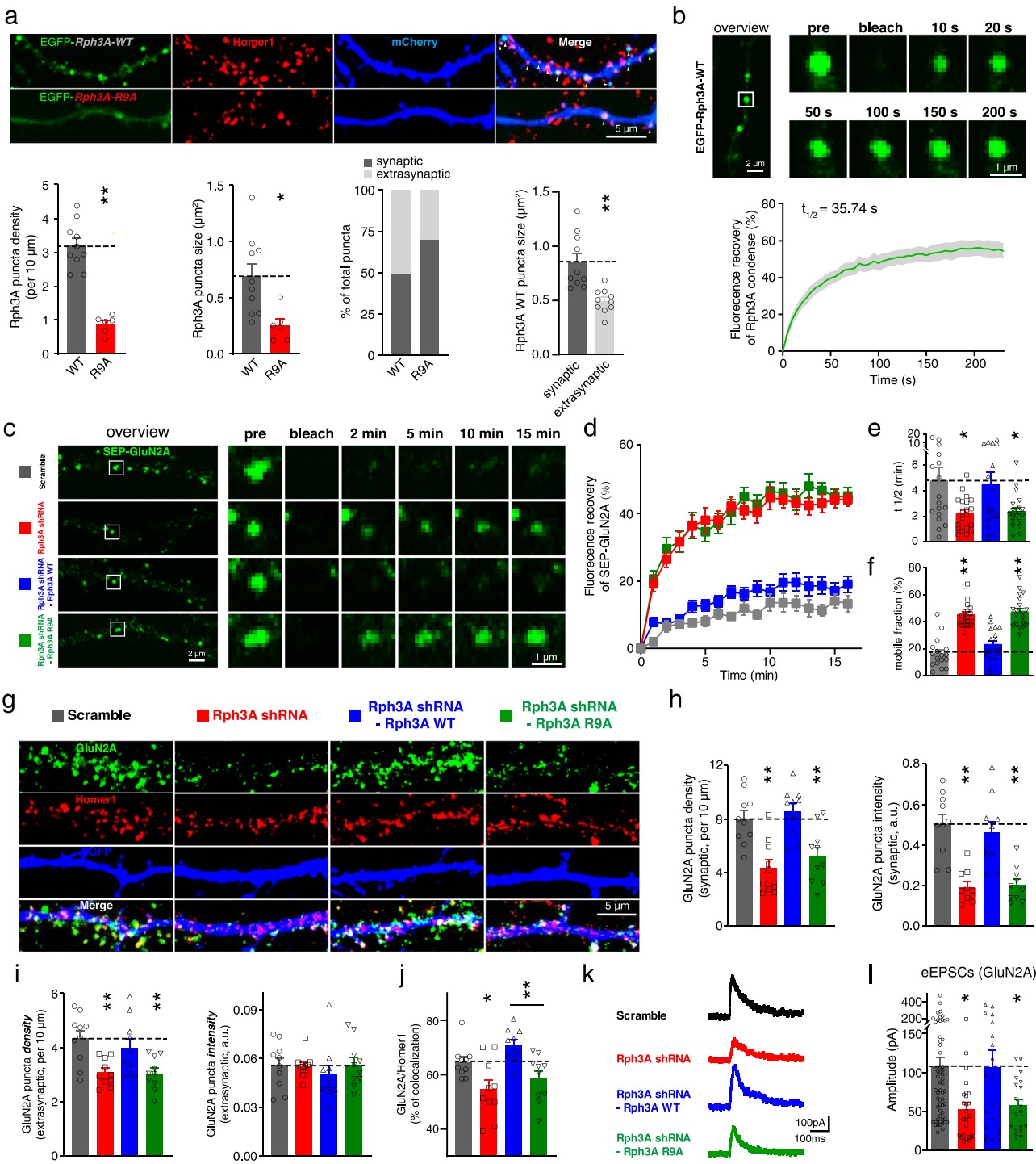

The suspensions were sonicated to release intercellular proteins. The lysates were then centrifuged at 18,000 × g for 50 min at 4 °C, and the cell debris was removed. The cleared lysates were loaded onto a Ni-NTA spin column (Thermo Scientific, 88229) preequilibrated with 5X volumes of Buffer A containing 10 mM imidazole. The columns were washed 7 times with 5X volumes of washing buffer (Buffer A containing 25 mM imidazole). The proteins were eluted with Buffer A containing 250 mM imidazole. The eluted protein-containing samples were loaded onto an Amicon Ultra-15 Centrifugal Filter (Millipore, UFC9010088) for concentration, and the buffer was replaced with Buffer B (50 mM Tris-HCl (pH 7.6), 150 mM NaCl, 1 mM EDTA, and 5% glycerol). The concentrations of the purified proteins were determined by measuring the UV absorbance at 280 nm. The proteins were then stored at −80 °C.

**In vitro droplet assay**

Recombinant fusion proteins were diluted to the appropriate concentration with Buffer B. The molecular crowding reagent PEG8000 (2%) was added to the protein solution. The protein solution was then immediately loaded into a homemade glass-bottom chamber. The chamber was imaged using an Olympus FV3000 microscope with a 100X objective. To capture droplet fusion events, images were taken every 3 s.

**FRAP assays**

FRAP assays were performed using an Olympus FV3000 microscope. Before bleaching, 3 images were captured to calculate the baseline fluorescence. For the optoDroplet assay, images were initially acquired

**Fig. 6 | Disruption of the phase separation capacity of Rph3A influenced the mobility, surface clustering, and synaptic localization of GluN2A in hippocampal neurons. a** Representative images of EGFP-tagged WT and R9A Rph3A at dendrites of hippocampal neurons. Yellow triangles indicate synaptic puncta, and white triangles indicate extrasynaptic puncta of Rph3A. The density, size, and synaptic localization of the puncta of WT and R9A Rph3A, and the sizes of synaptic and extrasynaptic WT Rph3A puncta were also quantified. The data are displayed as the mean ± SEM (WT: $n = 10$ dendrites from 5 neurons; R9A: $n = 6$ dendrites from 3 neurons, $**p < 0.01$, two-tailed unpaired $t$ test). **b** Representative images and quantification of fluorescence recovery from the FRAP analysis of Rph3A puncta at the dendrites of hippocampal neurons. The data are displayed as the mean ± SEM ($n = 11$ puncta). **c** Representative images from the FRAP assay of SEP-GluN2A in Rph3A-knockdown and Rph3A-reexpressing (WT and R9A) hippocampal neurons. **d** Quantification of the fluorescence recovery shown in **c**. The data are displayed as the mean ± SEM (Scramble: $n = 17$ puncta; Rph3A shRNA: 20 puncta; Rph3A shRNA-Rph3A WT: 18 puncta; Rph3A shRNA-Rph3A R9A: 20 puncta). **e** The recovery rate was calculated as the half-time of the recovery curve. The data are displayed as the mean ± SEM (n numbers are defined in d, $*p < 0.05$, one-way ANOVA followed by

Dunnett's multiple comparisons test). **f** The mobile fraction of GluN2A was calculated based on the plateau of the recovery curve. The data are displayed as the mean ± SEM (n numbers are defined in d, $**p < 0.01$, one-way ANOVA followed by Dunnett's multiple comparisons test). **g** Representative images of surface GluN2A and Homer1 staining in Rph3A-knockdown and Rph3A-reexpressing neurons. **h-j** Quantification of synaptic and extrasynaptic surface GluN2A cluster fluorescence intensity and density and the percentage of synaptic GluN2A clusters. The data are displayed as the mean ± SEM ($n = 10$ dendrites from 5 neurons for each group, $*p < 0.05$, $**p < 0.01$, one-way ANOVA followed by Dunnett's multiple comparisons test (h, i) or by Tukey's multiple comparisons test (j)). **k** Representative trace of NMDA eEPSCs in Rph3A-knockdown and Rph3A-reexpressing neurons. **l** Quantification of NMDA eEPSCs in Rph3A-knockdown and Rph3A-reexpressing neurons. The data are displayed as the mean ± SEM (Scramble: $n = 58$ neurons; Rph3A shRNA: 25 neurons; Rph3A shRNA-Rph3A WT: 22 neurons; Rph3A shRNA-Rph3A R9A: 19 neurons, $*p < 0.05$, one-way ANOVA followed by Dunnett's multiple comparisons test). The full images of **a**, **b**, **c**, **g** are showed in Supplementary Fig. 13. Source data and $p$ values of **a**, **b**, **d**, **e**, **f**, **h**, **i**, **j** and **l** are provided in the Source Data file.

every 1 s, and the droplets were then bleached using a 488-nm or 561-nm laser at maximum intensity for 1 s; after bleaching, images were acquired every 1 s for 1 min. For the in vitro droplet assay, images were initially acquired every 3 s, and bleaching was performed using a 488-nm or 561-nm laser at maximum intensity for 1.8 s; after bleaching, images were acquired every 3 s for 3 or 5 min. Images of neurons expressing Rph3A-EGFP were initially acquired every 5 s, and bleaching was performed using a 488-nm laser at maximum intensity for 0.43 s; after bleaching, images were acquired every 5 s for 4 min. Images of neurons expressing SEP-GluN2A/GluN2B were initially acquired every 60 s or 10 s, and bleaching was performed using a 488-nm laser at maximum intensity for 0.44 s; after bleaching, images were acquired every 60 s or 10 s for 16 min or 3 min, respectively. The obtained intensity recovery traces were normalized to the baseline fluorescence intensity by cellSens (V2.2) and fitted to a one-phase association exponential equation (Prism 5, GraphPad). The half-time was calculated as the time at which half of the fluorescence had recovered. The mobile fractions were determined according to the plateau of the fitting curve.

## Surface staining of GluN2A/GluN2B

Cultured neurons were washed with HBS 2 times and incubated with an anti-GluN2A/GluN2B antibody (Alomone AGC-002/AGC-003, 1:200 dilution) for 20 min at 37 °C in HBS buffer. After two washes with HBS, the cells were fixed in 4% PFA + 4% sucrose for 10 min on ice. After three washes in PBS for 5 min, the cells were blocked in blocking buffer (5% BSA and 2% goat serum in PBS) for 30 min at room temperature. Then, the cells were incubated with goat anti-mouse secondary antibodies (Abbkine A23610, 1:200 dilution) for 30 min at room temperature in the dark. After three washes in PBS for 5 min, the cells were fixed again in 4% PFA + 4% sucrose for 10 min. After three washes in PBS for 5 min, the cells were blocked in blocking buffer for 30 min at room temperature. Then, the cells were incubated with primary antibody of Homer1(Synaptic Systems 160011, 1:1000 dilution) or PSD95 (Neuromab 75-028, 1:2000 dilution) for overnight at 4 degrees. After three washes in PBS for 5 min, the cells were incubated with goat anti-rabbit secondary antibodies (Invitrogen A32732, 1:500 dilution) for 30 min at room temperature in the dark. Images were acquired using an Olympus FV3000 microscope with a 100X objective. The fluorescence intensity and density of the puncta revealed by antibody staining were quantified using ImageJ (1.53k).

## Co-IP experiments

For protein-protein interaction experiments, 20 μg of total plasmids including Flag-tagged Rph3A, myc-tagged GluN2A or HA-tagged PSD95 were transfected to a 10-cm dish. 48 h after transfection,

transfected dishes were washed twice with PBS. Cells were then collected in 700 μL of lysis buffer containing: 50 mM Tris-HCl, 1 mM EDTA, 150 mM NaCl, 1% Igepal CA-630. Anti-Flag or Anti-myc magnetic agarose beads (Life Technology) was washed six times with lysis buffer, and then added to 600 μL samples and rotated at 4 °C overnight. Then, the agarose beads were washed six times with lysis buffer, the proteins were eluted with sample buffer (Life Technology) for further western blot analysis. SDS-PAGE was performed by using NuPAGE gels (Life Technology). The membrane was scanned with an infrared imaging system (Odyssey). Monoclonal antibodies against Flag tag (Abmart: M20008L, 1:1000 dilution), myc tag (Abmart: M20002H, 1:1000 dilution) and HA tag (Abmart: M20003L, 1:1000 dilution) were used.

## Electrophysiology

Whole-cell voltage-clamp recordings were performed with an EPC10 patch-clamp amplifier (HEKA). Neurons plated on coverslips were maintained in an external solution (150 mM NaCl, 4 mM KCl, 2 mM CaCl₂, 1 mM MgCl₂, 10 mM HEPES, and 10 mM D-glucose, pH adjusted to 7.4, Osm: 315) during the recordings. Whole-cell patch-clamp recording was performed using 4–5 MΩ microelectrodes (World Precision Instruments). The internal solution contained 135 CsMeSO₄, 8 NaCl, 10 HEPES, 4 Mg-ATP, 0.3 Na-GTP, 0.3 EGTA, and 5 QX-314 (pH adjusted to 7.2, Osm: 305). Synaptic responses were evoked by 0.4-ms current injection (90-100 μA) delivered through a concentric bipolar electrode (CBBEB75, FHC, Bowdoin, ME, United States) with an isolated pulse stimulator (Model 2000, A-M Systems). GluN2A-containing NMDAR-mediated eEPSCs were pharmacologically isolated by the addition of 100 μM PTX, 10 μM NBQX, and 2 μM Ro25-6981 to the external solution. Paired pulse ratio was induced with two adjacent stimulus with 50 ms interval. Series resistance was compensated to 60–70%, and recordings with a series resistance of >20 MΩ were rejected. The data were analyzed using Clampfit 10.2 (Molecular Devices), Igor 4.0 (Wave Metrics) and Prism 8 (GraphPad Software Version 8.0.2 (263)).

## Statistics and reproducibility

Statistical tests were performed using Prism 8 (GraphPad Software Version 8.0.2 (263)) and mentioned in figure legends. All data are shown as the mean ± SEM calculated by Prism 8. All the experiments were repeated independently at least 3 times with similar results.

## Ethics statement

The mice were kept in a temperature- and relative humidity-controlled environment (22 ± 2 °C, 40–70%) with a 12-h light/dark cycle and free access to food and water. All animal studies were conducted according to the Guide for the Care and Use of Laboratory Animals (8th edition)

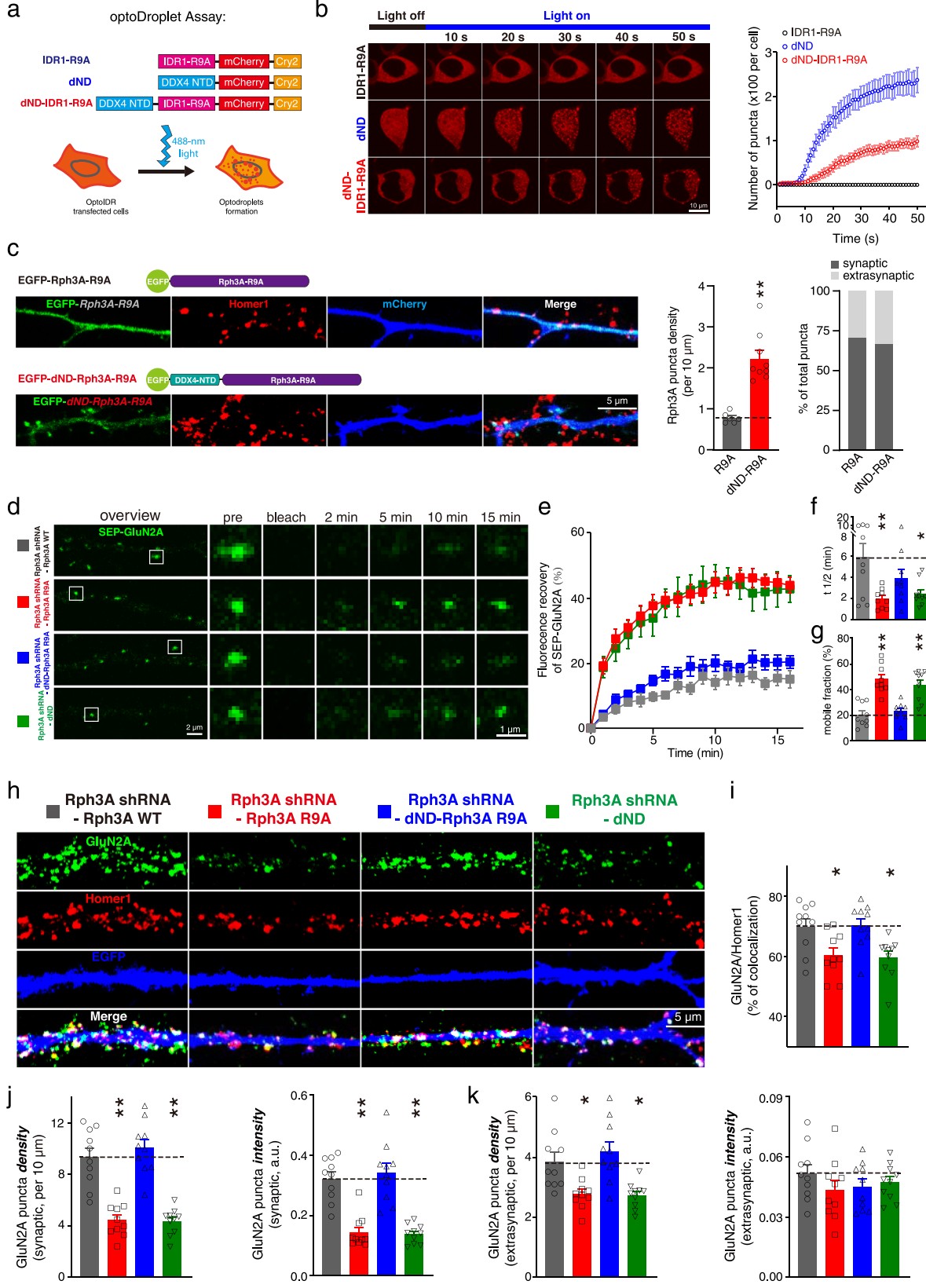

and approved by the Animal Experiments and Experimental Animal Welfare Committee of Capital Medical University (Approval ID: AEEI-2019-013).

**Reporting summary**

Further information on research design is available in the Nature Portfolio Reporting Summary linked to this article.

**Fig. 7 | Restoring phase separation of Rph3A reinstates its effect on surface clustering and mobility of GluN2A. a** Schematic illustration of dND-fused constructs used to restore phase separation of Rph3A. **b** OptoDroplet assay of indicated constructs in a. The number of puncta per cell was plotted. Data are displayed as the mean ± SEM (dND: $n = 7$ cells; dND-IDR1-R9A cells: $n = 9$ cells; IDR1-R9A: $n = 6$ cells). **c** Representative images of EGFP-tagged R9A and dND-fused Rph3A R9A at dendrites of hippocampus neurons. The density and synaptic localization of puncta were quantified. The data are displayed as the mean ± SEM (R9A: $n = 6$ dendrites from 3 neurons; dND-R9A: $n = 9$ dendrites from 5 neurons, $**p < 0.01$, two-tailed unpaired t test). **d** Representative images of the FRAP assay of SEP-GluN2A in Rph3A-knockdown and Rph3A-reexpressing neurons. **e** Quantification of the fluorescence recovery in d. The data are displayed as the mean ± SEM (Rph3A WT: $n = 9$ puncta; Rph3A R9A: $n = 9$ puncta; dND-Rph3A R9A: $n = 9$ puncta; dND: $n = 10$

puncta). **f** The recovery rate of SEP-GluN2A. The data are displayed as the mean ± SEM (n numbers are defined in e, $*p < 0.05$, $**p < 0.01$, one-way ANOVA followed by Tukey's multiple comparisons test). **g** The mobile fraction of SEP-GluN2A. The data are displayed as the mean ± SEM (n numbers are defined in e, $**p < 0.01$, one-way ANOVA followed by Tukey's multiple comparisons test). **h** Representative images of surface GluN2A and Homer1 staining in Rph3A-knockdown and Rph3A-reexpressing neurons. **i**–**k** Quantification of the percentage of synaptic GluN2A clusters, synaptic and extra-synaptic surface GluN2A cluster fluorescence intensity and density. The data are displayed as the mean ± SEM ($n = 10$ dendrites from 5 neurons for each group, $*p < 0.05$, $**p < 0.01$, one-way ANOVA followed by Tukey's multiple comparisons test). The full images of **c**, **d** and **h** are showed in Supplementary Fig. 13. Source data and p values of **b**, **c**, **e**, **f**, **g**, **i**, **j** and **k** are provided in the Source Data file.

## Data availability
The source data generated in this study are provided as a Source Data file with this paper. Source data are provided with this paper.

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

## Acknowledgements

This work was supported by grants from the National Science Foundation of China [81925011, 92149304] (C.Z.); the National Key R&D Program of China [2017YFA0105201] (C.Z.); the National Science Foundation of China [31900698, 32170954] (M.W.); the National Science Foundation of China [32100763] (L.Y.); the Key-Area Research and Development Program of Guangdong Province [2019B030335001] (C.Z.); the Youth Beijing Scholars Program (015) (C.Z.); the Support Project of High-Level Teachers in Beijing Municipal Universities [CIT&TCD20190334] (C.Z.); and the Beijing Advanced Innovation Center for Big Data-based Precision Medicine, Capital Medical University, Beijing, China [PXM2021_014226_000026] (C.Z.).

## Author contributions

L.Y., M.W., and C.Z. designed the research; L.Y., M.W., and Y.W. conducted the research and analyzed the data; J.Z., S.L., M.L., S.W., K.L., and Z.D. analyzed the data; and L.Y., M.W., and C.Z. wrote the paper.

## Competing interests

The authors declare no competing interests.
