## [Peer Review File · Nature Communications]

Rabphilin-3A undergoes phase separation to regulate GluN2A mobility and surface clusteringREVIEWER COMMENTS

Reviewer #1 (Remarks to the Author):

The manuscript by Yang, Wei, Wang and coworkers addresses a relevant topic in the field of glutamatergic neurotransmission, namely the characterization of the detailed molecular mechanisms involved in subunit specific regulation NMDA receptors. Starting from previous studies demonstrating that Rph3A regulates the surface retention of GluN2A-containing NMDA receptors, the authors found that phase separation of Rph3A plays a key role to condense with GluN2A subunit of NMDA receptor, decrease the mobility of GluN2A and maintain the surface clustering of GluN2A in hippocampal neurons. This is overall an intriguing study adding relevant information to the identification of GluN2A-mediated interactions at neuronal surface. The first part of the study based on optodroplet analysis and use of cell lines – in vitro assays is well-performed and clearly presented. Conversely, I've some concern about the second part of the study in primary hippocampal neurons that could be improved by additional experiments and controls to fully support the conclusions raised by the authors (see below for details).

As a general issue, several References in the Introduction should be updated. To make an example, GluN2A/GluN2B role in diseases such as Parkinson, namely the role of GluN2A/GluN2B ratio at synapses, has been deeply analysed in much more recent papers that the only one mentioned by the authors in the introduction. Similarly, even if the main focus of the present manuscript is the formation of Rph3A/GluN2A complex, only one out of different studies describing the physiological or pathological role of this molecular complex is mentioned.

A main concern is related to the specificity of the observed effects in primary neurons. It is well-known that Rph3A has several synaptic partners, both at the pre- and post-synaptic level. In particular, taking into account that Rph3A forms a ternary complex through a direct interaction with PSD-95 and GluN2A, a more careful evaluation of the effect of Arg mutations on Rph3A clustering with PSD-95 would be of great help for a better understanding of the molecular mechanism responsible for GluN2A surface retention. Do Rph3A mutant forms still bind PSD-95 or modify the dendritic distribution of PSD-95? Or, more relevant, do Rph3A mutants modify the formation of GluN2A/PSD-95 complex as Rph3A wt does? These questions are strictly related to another key issue that should be addressed related to the role of Rph3A in driving synaptic versus extrasynaptic distribution of GluN2A. The authors nicely demonstrate that Rph3A mutants have a different effect compared to Rph3A wt on GluN2A surface staining and GluN2A stability at the membranes, but they do not address the role of Rph3A phase separation in regulating the balance between synaptic and extrasynaptic GluN2A.

It is known that one the main events induced by Rph3A silencing is dendritic spine loss. Accordingly, the authors should also evaluate if the observed effects on GluN2A localization/stability is associated to morphological alterations at dendritic spine level. Does Rph3A mutant rescue synaptic loss induced by Rph3A silencing?

The manuscript contains a long paragraph in the discussion section about the putative effect of Rph3A

phase separation on pre-synaptic interactions. However, to exclude that the observed effect on GluN2A in neurons is correlated to alterations of pre-synaptic glutamate release associated to altered Rph3A/Rab3A interaction, some control experiment evaluating the presynaptic Rph3A activity should be performed.

Reviewer #2 (Remarks to the Author):

In this manuscript, the authors discovered that Rabphilin 3a (RPH3A), a previously identified GluN2A binding proteins, has a capacity to undergo phase separation in heterologous cells and in vitro. The authors further showed that the residues 1-91 of RPH3A, which is an intrinsically disordered domain (IDR1), is responsible for the phase separation of RPH3A. The authors identified that 9 Arg residues in the IDR1 are required for the phase separation of RPH3A. Substitution of the 9 Arg residues with Ala ("R9A") eliminated the phase separation capacity of the protein. The author further showed that the cytoplasmic domain of GluN2B (the detailed information of CTD is missing in line 476 of the manuscript) can be recruited to the RPH3A droplets in heterologous cells and in vitro, though the enrichment of GluN2B-CTD in the RPH3A droplets is very modest. The authors went on to provide evidence that the R9A mutant of RPH3A fails to rescue RPH3A-mediated GluN2A clustering in synapses in cultured hippocampal neurons.

The study provides evidence showing that RPH3A, possibly via phase separation with GluN2A, can specifically regulate GluN2A subunit-containing NMDA receptor clustering and anchoring in synapses. The finding suggests a molecular mechanism underlying subunit-specific synaptic clustering and removal mechanism for different NMDA receptors, which is an important topic in neuroscience. However, the manuscript contains several issues that must be addressed.

1. A key issue is that both the full-length RPH3A and its IDR1 require Cry2-mediated oligomerization for phase separation to occur in cells. In vitro, phase separation of the full-length RPH3A and its IDR1 requires addition of crowding reagent. GluN2A seems to be just a client. One might argue that RPH3A does not phase separate in physiological relevant conditions. Will it be possible that formation of the RPH3A/GluN2A/PSD-95 complex, as demonstrated in the study by Stanic et al (NC, 2015, Ref 32 in the manuscript), induces RPH3A phase separation.
2. In Fig. 4H, the authors showed that IDR2 is required for GluN2A recruitment into RPH3A droplets. However, the study by Stanic et al showed that RPH3A NTD (i.e., IDR1-ZF) is responsible for binding to GluN2A. The authors will need to reconcile this discrepancy.
3. The authors should also dissect which regions of GluN2A-CTD are required for its enrichment into the RPH3A droplets. Is the region between 1349-1389 mapped by Stanic et al necessary and sufficient for GluN2A-CTD to phase separate with RPH3A?

4. Would the RPH3A-R9A mutant change its GluN2A binding? This is an important experiment to perform, as this may allow the authors to dissociate phase separation of RPH3A from RPH3A's binding to GluN2A.
5. Fig. 5A: EGFP-RPH3A puncta in neurons are NOT spine localized. This is again different from the finding by Stanic et al. Why? How can RPH3A puncta in shafts affect GluN2A mobility in synapses (either synaptic or extra-synaptic)? Based on the finding in the manuscript, the two proteins do not meet each other in neurons. RPH3A is expected to binding PSD-95. Why RPH3A is not localized to synapse?
6. It will be valuable to test whether Rab3a may modulate phase separation of RPH3A or RPH3A/GluN2A complex.
7. Imaging figures throughout the manuscript, Fig. 5B&C and Figures S3&4 in particular: The resolutions of the images are very low. In certain cases, each punctum only has a few pixels, making quantifications challenging.
8. Fig. 4D: was the image acquired with or without blue light on? What is the very large punctum with both GluN2A and RPH3A co-enriched?
9. Fig. 4H: in the image without light on, what is the big punctum with both GluN2A and RPH3A co-enriched?

Reviewer #3 (Remarks to the Author):

The work from Yang et. al. is substantial work that addresses if the scaffolding protein RPH3A can promote clustering of N-methyl-D-aspartate receptors through liquid-liquid phase separation and, subsequently, the functional consequences of clustering in cells. The work used mix of in cell studies, including the optogenetics platform, and in vitro reconstitution of RPH3A to demonstrate this protein can undergo LLPS on its own and this is driven by a single IDR region found inside the protein. Mutation of nine Arginine residues in this IDR region abolishes LLPS of RPH3A and the GluN2A subunit of NMDAR co-localizes with clusters formed by RPH3A. Furthermore the mobility of GluN2A in dendrites as accessed by FRAP is significantly reduced in the presence of RPH3A in dendritic clusters. The data presented by the authors offers strong evidence that RPH3A has the capability to undergo LLPS with biochemical data correlating strongly the in cell optogenetics data as well as these droplets, both in vitro and in cell, showing all the hallmarks of liquid droplets. From what this reviewer can see the data appears to be sound and acquired properly.

The relationship between RPH3A droplets and GluN2A isn't as clear given the data provided. Is it that RPH3A droplets form and subsequently GluN2A is recruited to RPH3A driven droplets or is the association between RPH3A and GluN2A together promotes LLPS? Given that the optogenetics platform strongly drives association and forms RPH3A clusters in cells, I suggest the best way to go after this question is for the authors to do a concentration dependency using their in vitro platform. The novelty the manuscript attempts to address, at least from the perspective of the field of biomolecular condensates, is the functional consequences of RPH3A on NMDAR receptors through liquid-liquid phase separation. The FRAP and electrophysiology data do demonstrate that removal of RPH3A and introduction of RPH3A R9A does affect the mobility of GluN2A and GluN2A-based action potential.

The authors suggest that the lack of mobility of GluN2A was due to the formation of surface clusters through LLPS. Given the data they have provided, this reviewer believes a more accurate interpretation is a correlation between the LLPS and mobility/action potential. One possibility is RPH3A would make larger order oligomers which tend to undergo LLPS and it is recruitment of these oligomers to GluN2A which drives the functional difference of GluN2A from GluN2B irrespective of LLPS occurring. It is also not clear whether the perturbation shown only affects the propensity of RPH3A to undergo LLPS or may affect another function that is unknown at this time. The data shown does show a slower recovery from FRAP due to WT RPH3A and faster recovery attached to the RPH3A R9A mutant. One can say FRAP is at best a proxy for the actual molecular mobility and does not directly separate the effect of LLPS from the molecular clustering itself, which may also drive the lower mobility.

Further suggestions:

Line 158/Figure 3C/3H: The authors use droplet size as a proxy for the extent of LLPS. However this property is a convolution of several factors including total protein concentration, propensity to undergo LLPS and surface tension. The authors should do a concentration titration to examine the extent of LLPS in vitro.

Line 196-198/Figure 4H: The authors demonstrated how IDR2 is required for co-localization of GluN2A using optogenetics in cell. The argument for this requirement would be significantly strengthened if a correlative in vitro assay was conducted as well.

Responses to the reviewers' comments on Yang et al.,
" RPH3A undergoes phase separation to regulate GluN2A mobility
and surface clustering "

We greatly appreciate the overall positive responses from the reviewers and thank them for their thoughtful and helpful suggestions and criticisms. As detailed below, we have addressed all of their concerns by revising the manuscript and figures and performing additional experiments, which are described in the new version of the manuscript. We hope that the reviewers find the revised paper substantially improved and suitable for publication in *Nature Communications*.

Below, the reviewers' comments are in italics, and our responses and the changes made in the revised manuscript are in bold.

Reviewer Comments:

Reviewer #1:

The manuscript by Yang, Wei, Wang and coworkers addresses a relevant topic in the field of glutamatergic neurotransmission, namely the characterization of the detailed molecular mechanisms involved in subunit specific regulation NMDA receptors. Starting from previous studies demonstrating that Rph3A regulates the surface retention of GluN2A-containing NMDA receptors, the authors found that phase separation of Rph3A plays a key role to condense with GluN2A subunit of NMDA receptor, decrease the mobility of GluN2A and maintain the surface clustering of GluN2A in hippocampal neurons.

This is overall an intriguing study adding relevant information to the identification of GluN2A-mediated interactions at neuronal surface. The first part of the study based on optodroplet analysis and use of cell lines – in vitro assays is well-performed and clearly presented. Conversely, I've some concern about the second part of the study in primary hippocampal neurons that could be improved by additional experiments and controls to fully support the conclusions raised by the authors (see below for details).

Response:

We are grateful for the reviewer's positive comments, and thank the reviewers for pointing our weakness.

As a general issue, several References in the Introduction should be updated. To make an example, GluN2A/GluN2B role in diseases such as Parkinson, namely the role of GluN2A/GluN2B ratio at synapses, has been deeply analysed in much more recent papers than the only one mentioned by the authors in the introduction. Similarly, even if the main focus of the present manuscript is the formation of Rph3A/GluN2A complex, only one out of different studies describing the physiological or pathological role of this molecular complex is mentioned.

Response:

We thank the reviewer for noticing this weakness. To make a more informative introduction about the GluN2A/GluN2B role in diseases, we have added more information about the GluN2A/GluN2B ratio in Parkinson's disease and levodopa therapy in the revised manuscript in lines 57 to 61 with the following text: "Furthermore, the ratio of GluN2A/GluN2B subunits in NMDARs has been found to be altered in rat and primate models of Parkinson's disease and under levodopa therapy, which is associated with the development of dyskinesia (Dunah, Wang et al. 2000, Picconi, Gardoni et al. 2004, Hallett, Dunah et al. 2005, Paille, Picconi et al. 2010). Targeting GluN2A-containing NMDARs has been considered as an approach for reducing dyskinesia under levodopa therapy (Gardoni, Sgobio et al. 2012, Mellone, Stanic et al. 2015)". To clearly describing the role of the RPH3A/GluN2A complex in synaptic function and disease, we added some statements in lines 89 to 95 with the following text: "Disrupting the interactions between RPH3A and GluN2A and between RPH3A and PSD95 or knocking down RPH3A could suppress the surface expression and synaptic localization of GluN2A, the NMDAR-mediated current, and LTP induction and negatively impact cognition (Stanic, Carta et al. 2015, Franchini, Stanic et al. 2019). The synaptic localization of RPH3A and its interaction with the GluN2A subunit were found to be increased in a rat model of Parkinson's disease and levodopa-induced dyskinesia (Stanic, Mellone et al. 2017)". The relevant references have also been updated in revised manuscript.

A main concern is related to the specificity of the observed effects in primary neurons. It is well-known that Rph3A has several synaptic partners, both at the pre- and post-synaptic level. In particular, taking into account that Rph3A forms a ternary complex through a direct interaction with PSD-95 and GluN2A, a more careful evaluation of the effect of Arg mutations on Rph3A clustering with PSD-95 would be of great help for a better understanding of the molecular mechanism responsible for GluN2A surface retention. Do

Rph3A mutant forms still bind PSD-95 or modify the dendritic distribution of PSD-95? Or, more relevant, do Rph3A mutants modify the formation of GluN2A/PSD-95 complex as Rph3A wt does? These questions are strictly related to another key issue that should be addressed related to the role of Rph3A in driving synaptic versus extrasynaptic distribution of GluN2A. The authors nicely demonstrate that Rph3A mutants have a different effect compared to Rph3A wt on GluN2A surface staining and GluN2A stability at the membranes, but they do not address the role of Rph3A phase separation in regulating the balance between synaptic and extrasynaptic GluN2A.

Response:

We thank the reviewer for raising this excellent point. To examining the effect of RPH3A phase separation on the PSD95 dendritic cluster distribution and GluN2A/PSD95 complex formation, we performed additional experiments, the results of which are shown in line 336 to 364 and Figure S8 in the revised manuscript. First, we labeled PSD95 and surface GluN2A by antibody in RPH3A knockdown neurons and neurons in which WT RPH3A or R9A RPH3A expression was rescued. We quantified the intensity and density of PSD95 clusters, and the colocalization of GluN2A with PSD95. We found that density of PSD95 clusters was lower and the colocalization of GluN2A with PSD95 was weaker in R9A RPH3A-reeexpressing neurons compared with those in WT RPH3A-reeexpressing neurons (Figure S8B and G). Second, we performed a Co-IP assay of GluN2A-CTD with PSD95 in the presence of WT or R9A mutant RPH3A, and a Co-IP assay of PSD95 with WT or R9A mutant RPH3A. We found that the R9A mutation decreased the promotion effect of RPH3A on the GluN2A/PSD95 interaction (Figure S8I), and that R9A mutation did not affect the binding of RPH3A to PSD95 (Figure S8J). Thus, we inferred that the phase separation of RPH3A mediated by the Arg-rich motif promoted the dendritic clustering of PSD95 and the formation of GluN2A/PSD95 complex.

We also performed an additional experiment to examine the role of RPH3A phase separation in regulating the balance between synaptic and extrasynaptic GluN2A, the results of which are shown in line 321 to 331 and Figure 6G–J in the revised manuscript. In this experiment, we used Homer1 as a synaptic marker and quantified synaptic and extrasynaptic GluN2A clusters as well as the percentage of synaptic GluN2A in RPH3A knockdown neurons and neurons in which WT RPH3A or R9A RPH3A expression was rescued. We found that the R9A mutation could not rescue the impaired synaptic (Figure 6H) and extrasynaptic (Figure 6I) clustering of GluN2A, while WT RPH3A could fully rescue these deficits. The synaptic localization of

GluN2A in R9A RPH3A-reexpressing neurons was also weaker than that in WT RPH3A-reexpressing neurons (Figure 6J). These results suggested that although RPH3A phase separation maintained both synaptic and extrasynaptic GluN2A clustering, it was more critical for the synaptic clustering of GluN2A.

It is known that one the main events induced by Rph3A silencing is dendritic spine loss. Accordingly, the authors should also evaluate if the observed effects on GluN2A localization/stability is associated to morphological alterations at dendritic spine level. Does Rph3A mutant rescue synaptic loss induced by Rph3A silencing?

Response:

We agree with the reviewer and thus performed an additional experiment, the results of which are shown in line 365 to 372 and Figure S9. We monitored dendritic spine morphology by expressing mCherry in RPH3A-knockdown and RPH3A-reexpressing neurons and quantified the dendritic spine density of these neurons. We found that the R9A mutation did not rescue the spine loss induced by RPH3A silencing while WT RPH3A did (Figure S9). Thus, our findings suggest that the change in the phase separation of RPH3A was associated with spine density alterations in hippocampal neurons.

The manuscript contains a long paragraph in the discussion section about the putative effect of Rph3A phase separation on pre-synaptic interactions. However, to exclude that the observed effect on GluN2A in neurons is correlated to alterations of pre-synaptic glutamate release associated to altered Rph3A/Rab3A interaction, some control experiment evaluating the presynaptic Rph3A activity should be performed.

Response:

We thank the reviewer for raising this point. In all experiments performed in cultured hippocampal neurons, we used the calcium phosphate method to transfect only three to five neurons that were distantly spaced on a cover slip, which prevented synaptic transmission between the transfected neurons. Besides, to exclude the potential presynaptic effect of RPH3A, we performed an additional electrophysiology experiment to record the paired-pulse ratio of GluN2A-dependent eEPSCs in RPH3A-knockdown and RPH3A-reexpressing neurons, the results of which are shown in line 379 to 381 and Figure S10. We found no differences in the paired pulse ratio between the scramble, RPH3A knockdown, RPH3A WT and RPH3A

R9A rescue groups (Figure S10). This finding suggests that the presynaptic glutamate release associated with the transfected neurons was not altered, thus excluding that observed effect on postsynaptic GluN2A currents is correlated to alterations of presynaptic activity.

Reviewer #2:

In this manuscript, the authors discovered that Rabphilin 3a (RPH3A), a previously identified GluN2A binding proteins, has a capacity to undergo phase separation in heterologous cells and in vitro. The authors further showed that the residues 1-91 of RPH3A, which is an intrinsically disordered domain (IDR1), is responsible for the phase separation of RPH3A. The authors identified that 9 Arg residues in the IDR1 are required for the phase separation of RPH3A. Substitution of the 9 Arg residues with Ala ("R9A") eliminated the phase separation capacity of the protein. The author further showed that the cytoplasmic domain of GluN2B (the detailed information of CTD is missing in line 476 of the manuscript) can be recruited to the RPH3A droplets in heterologous cells and in vitro, though the enrichment of GluN2B-CTD in the RPH3A droplets is very modest. The authors went on to provide evidence that the R9A mutant of RPH3A fails to rescue RPH3A-mediated GluN2A clustering in synapses in cultured hippocampal neurons.

Response:

We thank the reviewer for pointing out this omission about the detailed information of GluN2A CTD. We have provided the amino acid residues included in GluN2A-CTD, which is aa 1244–1464, in the Results: line 97 to 98, Methods: line 805 and Figure S3.

The study provides evidence showing that RPH3A, possibly via phase separation with GluN2A, can specifically regulate GluN2A subunit-containing NMDA receptor clustering and anchoring in synapses. The finding suggests a molecular mechanism underlying subunit-specific synaptic clustering and removal mechanism for different NMDA receptors, which is an important topic in neuroscience. However, the manuscript contains several issues that must be addressed.

1. A key issue is that both the full-length RPH3A and its IDR1 require Cry2-mediated oligomerization for phase separation to occur in cells. In vitro, phase separation of the full-length RPH3A and its IDR1 requires addition of crowding reagent. GluN2A seems to be just a client. One might argue that RPH3A does not phase separate in physiological relevant conditions. Will it be possible that formation of the RPH3A/GluN2A/PSD-95 complex, as demonstrated in the study by Stanic et al (NC, 2015, Ref 32 in the manuscript), induces RPH3A phase separation.

Response:

We thank the reviewer for raising this great point. Protein concentration is an important factor that determines protein phase separation (Chen, Wu et al. 2020). Based on previously described approaches (Milovanovic, Wu et al. 2018, Sabari, Dall'Agnese et al. 2018), blue light-activated Cry2 oligomerization and a crowding reagent (PEG 8000) were used to increase the local concentration of the proteins in HEK293 cells and in a cell-free system to induce protein phase separation. To further address the reviewer's concern, we concentrated purified EGFP-tagged RPH3A to 50 μ M in buffer containing a physiological salt concentration (150 mM NaCl) without crowding reagent. We could observe droplet formation of RPH3A in this condition (Figure 3A). Our results suggests that RPH3A alone could undergo phase separation when present at the appropriate concentration.

As the reviewer suggested, to examined whether GluN2A and PSD95 induced the phase separation of RPH3A, we performed additional experiments, the results of which are shown in line 245 to 273 and Figure 5, S4 and S5 in the revised manuscript. First, we expressed different combinations of EGFP-tagged RPH3A, mCherry-tagged GluN2A-CTD, and BFP-tagged PSD95 in HEK293 cells. We found that RPH3A, GluN2A-CTD, and PSD95 remained diffuse in cells when they were individually expressed, that coexpressing GluN2A-CTD and PSD95 induced puncta formation in HEK293 cells and that RPH3A was recruited to the puncta composed of GluN2A-CTD and PSD95, resulting in the formation of larger puncta (Figure 5A and B). R9A mutant RPH3A was also recruited to the puncta; however, R9A mutant RPH3A recruitment did not change puncta size (Figure 5A and B). A FRAP assay revealed that the RPH3A condensates showed liquid-like properties (Figure S4). Second, we purified the recombinant EGFP-RPH3A, mCherry-GluN2A-CTD, and PSD95-BFP fusion proteins and tested the droplet formation of mixtures of different combinations of proteins. In the cell-free system, the mixture of RPH3A, GluN2A and PSD95 formed droplets that were larger than those composed of only RPH3A (Figure 5C and D). Increasing the concentration of GluN2A/PSD95 complex induced larger droplets of RPH3A (Figure S5C). Neither GluN2A nor PSD95 alone promoted the phase separation of RPH3A in HEK293 cells or in the cell-free system (Figure 5, S5A and S5B). These results suggested that the GluN2A/PSD95 complex promoted the phase separation of RPH3A.

2. In Fig. 4H, the authors showed that IDR2 is required for GluN2A recruitment into RPH3A

droplets. However, the study by Stanic et al showed that RPH3A NTD (i.e., IDR1-ZF) is responsible for binding to GluN2A. The authors will need to reconcile this discrepancy.

Response:

We appreciate the reviewer for pointing out this discrepancy. To address this issue, we performed additional experiments, the results of which are shown in line 224 to 233 and Figure S2 in the revised manuscript. First, we tested whether N-terminal aa 1–179 of RPH3A (containing the IDR1-ZF sequence (aa 1–157)), a region reported to interact with GluN2A-CTD (Stanic, Carta et al. 2015), cocondensed with GluN2A-CTD in the optoDroplet assay. We still did not observe any cocondensation (Figure S2A and S2B), which suggested that the aa 158-179 is not responsible for the discrepancy. Second, we performed Co-IP assays of GluN2A-CTD with different truncated RPH3A constructs, including NTD, 1-179 aa, IDR1+ZF, IDR2+CTD and CTD of RPH3A (Figure S2C), and found that both IDR1+ZF and IDR2 interacted with GluN2A-CTD, and the lack of either IDR1+ZF or IDR2 weakened the interaction between GluN2A-CTD and RPH3A (Figure S2D and E). Therefore, we inferred that both IDR1+ZF and IDR2 were required for the strong interaction between GluN2A-CTD and RPH3A, which is responsible for cocondensation in the optoDroplet assay.

3. The authors should also dissect which regions of GluN2A-CTD are required for its enrichment into the RPH3A droplets. Is the region between 1349-1389 mapped by Stanic et al necessary and sufficient for GluN2A-CTD to phase separate with RPH3A?

Response:

We agree with the reviewer. We performed additional experiment to address this issue, the results of which are shown in line 234 to 243 and Figure S3. We tested the condensation of RPH3A with different truncated GluN2A-CTD constructs including aa 1244–1348, aa 1244–1388, aa 1349–1388, aa 1349–1464 and aa 1389–1464 of GluN2A-CTD, as illustrated in Figure S3A. We observed no cocondensation of RPH3A with any of the truncated GluN2A-CTD constructs (Figure S3B–G). We then performed a Co-IP assay of different truncated GluN2A-CTD constructs with RPH3A and found that all the truncated GluN2A-CTD constructs showed weaker interaction with RPH3A compared with the FL GluN2A-CTD (Figures S3H and I). Therefore, these results indicated that the sequence from aa 1244 to the C-terminus (aa 1464) of

GluN2A was necessary and sufficient for GluN2A-CTD to undergo phase separation with RPH3A.

4. Would the RPH3A-R9A mutant change its GluN2A binding? This is an important experiment to perform, as this may allow the authors to dissociate phase separation of RPH3A from RPH3A's binding to GluN2A.

Response:

We thank the reviewer for raising this excellent point. To address this question, we performed an additional Co-IP assay of WT and R9A mutant RPH3A with GluN2A-CTD (Figure S2D and E), and we found no significant change in GluN2A-CTD binding with the R9A mutant RPH3A compared with WT RPH3A. As RPH3A has been reported to interact with PSD95 (Stanic, Carta et al. 2015), we also performed an additional Co-IP assay of WT and R9A mutant RPH3A with PSD95 and found no change in the binding of the R9A mutant to PSD95 compared with WT RPH3A (Figure 8J). Thus, we believe that these results distinguish the role of the Arg-rich motif in phase separation from a role in binding GluN2A or PSD95, indicating that R9A mutant RPH3A was an appropriate construct for examining the function of RPH3A phase separation.

5. Fig. 5A: EGFP-RPH3A puncta in neurons are NOT spine localized. This is again different from the finding by Stanic et al. Why? How can RPH3A puncta in shafts affect GluN2A mobility in synapses (either synaptic or extra-synaptic)? Based on the finding in the manuscript, the two proteins do not meet each other in neurons. RPH3A is expected to binding PSD-95. Why RPH3A is not localized to synapse?

Response:

We acknowledge the incorrect use of the C-terminally fused EGFP-tagged RPH3A construct in the previous Figure 5A. As the PSD95-binding motif is located at the C-terminus of RPH3A, it may influence the binding of PSD95 and RPH3A and subsequently affect the synaptic localization of RPH3A. To address this issue, we performed additional experiments, the results of which are shown in line 280 to 285 and Figure 6A. We transfected neurons with EGFP-tagged RPH3A fused to the N-terminus and improved the experimental design by labeling synapses by antibody against Homer1 (Figure 6A). Besides, confocal images were obtained with a 100X objective instead of a 60X objective to improve the resolution. We quantified the

percentage of GluN2A colocalized with Homer1 and found that nearly 50% of the WT RPH3A puncta were synaptically localized (Figure 6A). It has also been shown that RPH3A localizes to dendritic spines and shafts (Stanic, Carta et al. 2015) and that GluN2A is also found in dendrite shafts (Petralia, Wang et al. 2010). Therefore, we think that it makes sense that RPH3A puncta are located at extrasynaptic sites and regulate the mobility and clustering of extrasynaptic GluN2A.

To address how phase separation of RPH3A regulate synaptic and extrasynaptic GluN2A, we performed additional experiments, the results of which are shown in line 321 to 331 and Figure 6G–J. We used Homer1 as a synaptic marker and quantified synaptic and extrasynaptic GluN2A clusters as well as the percentage of synaptic GluN2A in RPH3A knockdown neurons and WT/R9A RPH3A reexpressing neurons. We found that the knockdown of RPH3A decreased the density of both synaptic and extrasynaptic surface GluN2A clusters, and the fluorescence intensity of synaptic GluN2A clusters was decreased in the RPH3A knockdown group compared with the control group. WT RPH3A could rescue these phenotypes, while R9A mutant RPH3A could not (Figure 6H and 6I). The synaptic localization of GluN2A in WT RPH3A-reexpressing group was stronger than that in R9A RPH3A-reexpressing group (Figure 6J). These results revealed that the phase separation of RPH3A maintained both the synaptic and extrasynaptic surface GluN2A clustering and promoted the synaptic localization of GluN2A.

6. It will be valuable to test whether Rab3a may modulate phase separation of RPH3A or RPH3A/GluN2A complex.

Response:

We agree with the reviewer, and we therefore tested whether Rab3a modulates the phase separation of RPH3A in an optoDroplet assay (Figure S11). We co-transfected Rab3a with RPH3A-Cry2 in HEK293 cell and found that Rab3a coexpression fully abolished RPH3A droplet formation (Figure S11A). Rab3a has been reported to bind to the Rab-binding domain (i.e., IDR1+ZF) in the N-terminus of RPH3A (Stahl, Chou et al. 1996). However, IDR1, which did not bind to Rab3a in previous work (Stahl, Chou et al. 1996), formed droplets in the presence of Rab3a (Figure S11B). These results suggest that Rab3a binding impairs the phase separation of RPH3A.

To address whether Rab3a modulate RPH3A/GluN2A complex, we performed a Co-IP assay of GluN2A-CTD with RPH3A in the presence or absence of Rab3a (Figure S11C). We found that Rab3a decreased the interaction between GluN2A-CTD and RPH3A (Figure S11D). To date, we have not determined the physiological role of the suppression of Rab3a in RPH3A phase separation and the RPH3A-GluN2A interaction. We believe it will be valuable to examine this issue in future work.

7. Imaging figures throughout the manuscript, Fig. 5B&C and Figures S3&4 in particular: The resolutions of the images are very low. In certain cases, each punctum only has a few pixels, making quantifications challenging.

Response:

We appreciate the reviewer for pointing out this weakness. To address the reviewer's concern about the resolution of images of cultured neurons, we conducted the experiments again, using a 100X objective instead of a 60X objective to improve the resolution of all the images of cultured neuron and the precision of the quantification, as shown in Figure 6, Figure 7, Figure S6, Figure S8 and Figure S9 in revised manuscript. We hope the reviewer find the quality of our imaging to be improved better for quantifications.

8. Fig. 4D: was the image acquired with or without blue light on? What is the very large punctum with both GluN2A and RPH3A co-enriched?

Response:

We apologize for the confusion in the previous submission and thank the reviewer for noticing it. The image in Figure 4D was acquired as time-lapse images of transfected HEK293 cell with blue light on. We observed that nearly 20% of the HEK293 cells showed puncta of different sizes in the optoDroplet assay without blue light exposure (Figure S13). It has been reported that the optoDroplet construct might occasionally be preactivated (Kilic, Lezaja et al. 2019, Zhang, Vigers et al. 2020). We assumed that the punctum might be due to the occasional preactivation of RPH3A-mch-Cry2 and that GluN2A was enriched in the punctum. We have added the description of the large punctums in Figure 4D in the revised manuscript.

9. Fig. 4H: in the image without light on, what is the big punctum with both GluN2A and RPH3A co-enriched?

Response:

We apologize for the confusion in the previous submission and thank the reviewer for noticing it. As stated above, we believe that this punctum resulted from the occasional preactivation of RPH3A-mch-Cry2. We have added the description of the large punctums in Figure 4H in the revised manuscript.

Reviewer #3

The work from Yang et. al. is substantial work that addresses if the scaffolding protein RPH3A can promote clustering of N-methyl-D-aspartate receptors through liquid-liquid phase separation and, subsequently, the functional consequences of clustering in cells. The work used mix of in cell studies, including the optogenetics platform, and in vitro reconstitution of RPH3A to demonstrate this protein can undergo LLPS on its own and this is driven by a single IDR region found inside the protein. Mutation of nine Arginine residues in this IDR region abolishes LLPS of RPH3A and the GluN2A subunit of NMDAR co-localizes with clusters formed by RPH3A. Furthermore the mobility of GluN2A in dendrites as accessed by FRAP is significantly reduced in the presence of RPH3A in dendritic clusters.

The data presented by the authors offers strong evidence that RPH3A has the capability to undergo LLPS with biochemical data correlating strongly the in cell optogenetics data as well as these droplets, both in vitro and in cell, showing all the hallmarks of liquid droplets. From what this reviewer can see the data appears to be sound and acquired properly.

Response:

We are grateful for the reviewer's positive comments – it is very nice to have the reviewer's comments like “strong evidence”, and to have one's data called "sound" and "acquired properly".

The relationship between RPH3A droplets and GluN2A isn't as clear given the data provided. Is it that RPH3A droplets form and subsequently GluN2A is recruited to RPH3A driven droplets or is the association between RPH3A and GluN2A together promotes LLPS? Given that the optogenetics platform strongly drives association and forms RPH3A clusters in cells, I suggest the best way to go after this question is for the authors to do a concentration dependency using their in vitro platform. The novelty the manuscript attempts to address, at least from the perspective of the field of biomolecular condensates, is the functional consequences of RPH3A on NMDAR receptors through liquid-liquid phase separation. The FRAP and electrophysiology data do demonstrate that removal of RPH3A and introduction of RPH3A R9A does affect the mobility of GluN2A and GluN2A-based

action potential.

Response:

We thank the reviewer for raising this point. To address this issue, we performed additional experiments, the results of which are shown in line 178 to 181, 263 to 273 and Figure 3A, Figure 5C, 5D and Figure S5 in the revised manuscript. First, we concentrated purified EGFP-tagged RPH3A to 50 μ M in buffer containing a physiological salt concentration (150 mM NaCl) without crowding reagent and tested the droplet formation. Our results provided evidence that RPH3A can undergo phase separation by itself in a cell-free system without a crowding reagent (Figure 3A), which suggested that RPH3A alone could undergo phase separation at the appropriate concentration. Second, as the reviewer suggested, we used the in vitro platform to examine whether GluN2A and PSD95, a scaffolding protein that binds to GluN2A and RPH3A (Stanic, Carta et al. 2015), promoted the phase separation of RPH3A. We purified the recombinant EGFP-RPH3A, mCherry-GluN2A-CTD, and PSD95-BFP fusion proteins and tested the droplet formation of mixtures of different combinations of proteins. We found that either GluN2A-CTD or PSD95 alone could only be recruited to RPH3A droplets, and they did not influence RPH3A droplet size in the cell-free system (Figure 5C and D). However, mixing RPH3A with GluN2A-CTD and PSD95 induced the formation of droplets that were larger than those composed of RPH3A alone (Figure 5C and 5D). We increased the concentrations of GluN2A-CTD (10 μ M, 20 μ M and 30 μ M), PSD95 (10 μ M, 20 μ M and 30 μ M) and the GluN2A-CTD/PSD95 complex (10 μ M and 20 μ M) in the mixture and found that only an increased concentration of the GluN2A-CTD/PSD95 complex induced the formation of larger droplets (Figure S5). Our findings from the cell-free system suggest that RPH3A alone could form droplets and the GluN2A-CTD/PSD95 complex promotes the phase separation of RPH3A in a concentration-dependent manner.

In addition, we expressed different combinations of EGFP-tagged RPH3A, mCherry-tagged GluN2A-CTD, and BFP-tagged PSD95 in HEK293 cells and observed the distribution of the proteins in HEK293 cells, the results of which are shown in line 249 to 262 and Figure 5A, 5B and S4. We found that RPH3A, GluN2A-CTD, and PSD95 remained diffuse in cells when they were individually expressed (Figure 5A). The coexpression of GluN2A-CTD with PSD95 induced puncta to form in HEK293 cells (Figure 5A). WT RPH3A was recruited to GluN2A-CTD/PSD95 puncta and induced the formation of larger puncta, while R9A RPH3A recruitment did not increase puncta size (Figure 5A and B). A FRAP assay revealed that the RPH3A condensates

showed liquid-like properties (Figure S4). Neither GluN2A nor PSD95 alone promoted the phase separation of RPH3A in HEK293 cells (Figure 5A and B). These results suggested that the GluN2A-CTD/PSD95 complex in cells could recruit RPH3A and further promote the phase separation of RPH3A.

The authors suggest that the lack of mobility of GluN2A was due to the formation of surface clusters through LLPS. Given the data they have provided, this reviewer believes a more accurate interpretation is a correlation between the LLPS and mobility/action potential. One possibility is RPH3A would make larger order oligomers which tend to undergo LLPS and it is recruitment of these oligomers to GluN2A which drives the functional difference of GluN2A from GluN2B irrespective of LLPS occurring. It is also not clear whether the perturbation shown only affects the propensity of RPH3A to undergo LLPS or may affect another function that is unknown at this time. The data shown does show a slower recovery from FRAP due to WT RPH3A and faster recovery attached to the RPH3A R9A mutant. One can say FRAP is at best a proxy for the actual molecular mobility and does not directly separate the effect of LLPS from the molecular clustering itself, which may also drive the lower mobility.

Response:

As stated above, we found that GluN2A/PSD95 puncta could recruit RPH3A in HEK293 cells and GluN2A/PSD95 complex could promote the phase separation of RPH3A (Figure 5). We agreed that the GluN2A or PSD95 could recruit RPH3A in the neuronal surface. We preferred to consider LLPS occurring due to that we observed that EGFP-tagged WT RPH3A formed spherical puncta with liquid properties in neuronal dendrites, while R9A RPH3A tended to remain diffuse (Figure 6A and B).

To furtherly validate that the phase separation of RPH3A modulate surface clustering and stability of GluN2A in hippocampal neurons, we restored the phase separation of RPH3A R9A by fusing the disordered N-terminus of DDX4 (aa 1-236) (dND), which was reported to drive liquid-liquid phase separation (Nott, Petsalaki et al. 2015), to the N-terminus of RPH3A R9A, and quantify the mobility and surface clustering of GluN2A by FRAP assay of SEP-GluN2A and immunostaining of surface GluN2A, the results of which are shown in line 386 to 408 and Figure 7. We found that the recovery rate and mobile fraction of SEP-GluN2A in dND fused RPH3A R9A group were comparable to those in RPH3A WT group, while those in RPH3A R9A and dND groups were not (Figure 7D–G). The density of synaptic and extrasynaptic clusters, the intensity of synaptic GluN2A clusters and the synaptic localization of

GluN2A in dND fused RPH3A R9A group were comparable to those in RPH3A WT group, while those in RPH3A R9A and dND were not (Figure 7H–K). Thus, our results indicated that phase separation is critical for RPH3A to maintain the clustering and mobility of GluN2A in hippocampal neuron.

Several results described in the revised manuscript suggested that the R9A mutant RPH3A showed normal expression and functions (except for phase separation): 1) The expression level of R9A mutant RPH3A was comparable to that of the WT RPH3A in HEK293 cells (Figures S2D, S8I and S8J). 2) The interactions between RPH3A and GluN2A or PSD95 did not differ when the R9A mutant was used (Figures S2D and S8J). 3) R9A RPH3A could form a ternary complex with GluN2A and PSD95 (Figure 5A) and promoted the GluN2A/PSD95 interaction (Figure S8I). Although we could not exclude the possibility of other unknown influences of the R9A mutant, we thought that the R9A mutant was a suitable construct for disrupting the phase separation of RPH3A.

To further confirm the effect of RPH3A phase separation on the mobility of GluN2A, we performed FRAP assay of GluN2A in HEK293 cells expressing BFP-tagged PSD95, EGFP-tagged GluN2A-CTD, and mCherry-tagged RPH3A, the results of which are shown in line 306 to 314 and Figure S7. We found that compared with that in the mCherry group and the RPH3A R9A group, the EGFP fluorescence recovery rate was significantly slower in the RPH3A WT group (Figure S7E). The proportion of EGFP fluorescence recovery was also lower in the RPH3A WT group (Figure S7F). Therefore, we inferred that there is an association between the lower mobility of GluN2A and the phase separation of RPH3A.

Further suggestions:

Line 158/Figure 3C/3H: The authors use droplet size as a proxy for the extent of LLPS. However this property is a convolution of several factors including total protein concentration, propensity to undergo LLPS and surface tension. The authors should do a concentration titration to examine the extent of LLPS in vitro.

Response:

We agree with the reviewer and thus performed a concentration titration of purified WT and truncated RPH3A proteins, the results of which are shown in line 181 to 183, 190 to 192 and Figure 3B and G. We found that the droplets size decreased with decreasing protein concentration (20 μ M, 10 μ M, 5 μ M and 2 μ M in Figure 3B) (20 μ M, 10 μ M and 5 μ M in Figure 3G). Considering that the salt concentration also regulates

protein phase separation, we also examined the extent of RPH3A phase separation in buffers containing different salt concentrations (150, 200, 250 and 300 mM NaCl), the results of which are shown in line 181 to 183 and Figure 3C. We found that droplets size decreased significantly with increasing salt concentration (Figure 3C). These results indicated that RPH3A undergoes phase separation in a protein and salt concentration-dependent manner.

Line 196-198/Figure 4H: The authors demonstrated how IDR2 is required for co-localization of GluN2A using optogenetics in cell. The argument for this requirement would be significantly strengthened if a correlative in vitro assay was conducted as well.

Response:

We thank the reviewer for raising this point. In the revised manuscript, we performed Co-IP assays of different truncated RPH3A constructs with GluN2A-CTD, including NTD, 1-179 aa, IDR1+ZF, IDR2+CTD and CTD of RPH3A, the results of which are shown in line 228 to 231 and Figure S3C–3E. We found that in addition to IDR1+ZF, IDR2 interacted with GluN2A-CTD, which was not mentioned in the previous work (Stanic, Carta et al. 2015). Ablation of IDR2 significantly weakened the interaction of RPH3A with GluN2A-CTD. Therefore, we corrected the description of our finding as follows: “Together, our results suggested that in addition to aa 1–179 of RPH3A, IDR2 interacted with GluN2A-CTD, and the deletion of IDR2 impaired the interaction and condensation of RPH3A with GluN2A-CTD.” in line 231 to 233.

References

Chen, X., X. Wu, H. Wu and M. Zhang (2020). "Phase separation at the synapse." Nat Neurosci **23**(3): 301-310.

Dunah, A. W., Y. Wang, R. P. Yasuda, K. Kameyama, R. L. Huganir, B. B. Wolfe and D. G. Standaert (2000). "Alterations in subunit expression, composition, and phosphorylation of striatal N-methyl-D-aspartate glutamate receptors in a rat 6-hydroxydopamine model of Parkinson's disease." Mol Pharmacol **57**(2): 342-352.

Franchini, L., J. Stanic, L. Ponzoni, M. Mellone, N. Carrano, S. Musardo, E. Zianni, G. Olivero, E. Marcello, A. Pittaluga, M. Sala, C. Bellone, C. Racca, M. Di Luca and F. Gardoni (2019). "Linking NMDA Receptor Synaptic Retention to Synaptic Plasticity and Cognition." iScience **19**: 927-939.

Gardoni, F., C. Sgobio, V. Pendolino, P. Calabresi, M. Di Luca and B. Picconi (2012). "Targeting NR2A-containing NMDA receptors reduces L-DOPA-induced dyskinesias." Neurobiol Aging **33**(9): 2138-2144.

Hallett, P. J., A. W. Dunah, P. Ravenscroft, S. Zhou, E. Bezard, A. R. Crossman, J. M. Brotchie and D. G. Standaert (2005). "Alterations of striatal NMDA receptor subunits associated with the development of dyskinesia in the MPTP-lesioned primate model of Parkinson's disease." Neuropharmacology **48**(4): 503-516.

Kilic, S., A. Lezaja, M. Gatti, E. Bianco, J. Michelena, R. Imhof and M. Altmeyer (2019). "Phase separation of 53BP1 determines liquid-like behavior of DNA repair compartments." EMBO J **38**(16): e101379.

Mellone, M., J. Stanic, L. F. Hernandez, E. Iglesias, E. Zianni, A. Longhi, A. Prigent, B. Picconi, P. Calabresi, E. C. Hirsch, J. A. Obeso, M. Di Luca and F. Gardoni (2015). "NMDA receptor GluN2A/GluN2B subunit ratio as synaptic trait of levodopa-induced dyskinesias: from experimental models to patients." Front Cell Neurosci **9**: 245.

Milovanovic, D., Y. Wu, X. Bian and P. De Camilli (2018). "A liquid phase of synapsin and lipid vesicles." Science **361**(6402): 604-607.

Nagerl, U. V., N. Eberhorn, S. B. Cambridge and T. Bonhoeffer (2004). "Bidirectional activity-dependent morphological plasticity in hippocampal neurons." Neuron **44**(5): 759-767.

Nott, T. J., E. Petsalaki, P. Farber, D. Jervis, E. Fussner, A. Plochowitz, T. D. Craggs, D. P. Bazett-Jones, T. Pawson, J. D. Forman-Kay and A. J. Baldwin (2015). "Phase transition of a disordered nuage protein generates environmentally responsive membraneless organelles." Mol Cell **57**(5): 936-947.

Paille, V., B. Picconi, V. Bagetta, V. Ghiglieri, C. Sgobio, M. Di Filippo, M. T. Viscomi, C. Giampa, F. R. Fusco, F. Gardoni, G. Bernardi, P. Greengard, M. Di Luca and P. Calabresi (2010). "Distinct levels of dopamine denervation differentially alter striatal synaptic plasticity and NMDA receptor subunit composition." J Neurosci **30**(42): 14182-14193.

Petralia, R. S., Y. X. Wang, F. Hua, Z. Yi, A. Zhou, L. Ge, F. A. Stephenson and R. J. Wenthold (2010).

"Organization of NMDA receptors at extrasynaptic locations." Neuroscience **167**(1): 68-87.

Picconi, B., F. Gardoni, D. Centonze, D. Mauceri, M. A. Cenci, G. Bernardi, P. Calabresi and M. Di Luca (2004). "Abnormal Ca²⁺-calmodulin-dependent protein kinase II function mediates synaptic and motor deficits in experimental parkinsonism." J Neurosci **24**(23): 5283-5291.

Sabari, B. R., A. Dall'Agnesse, A. Boija, I. A. Klein, E. L. Coffey, K. Shrinivas, B. J. Abraham, N. M. Hannett, A. V. Zamudio, J. C. Manteiga, C. H. Li, Y. E. Guo, D. S. Day, J. Schuijers, E. Vasile, S. Malik, D. Hnisz, T. I. Lee, Cisse, II, R. G. Roeder, P. A. Sharp, A. K. Chakraborty and R. A. Young (2018). "Coactivator condensation at super-enhancers links phase separation and gene control." Science **361**(6400).

Stahl, B., J. H. Chou, C. Li, T. C. Sudhof and R. Jahn (1996). "Rab3 reversibly recruits rabphilin to synaptic vesicles by a mechanism analogous to raf recruitment by ras." EMBO J **15**(8): 1799-1809.

Stanic, J., M. Carta, I. Eberini, S. Pelucchi, E. Marcello, A. A. Genazzani, C. Racca, C. Mülle, M. Di Luca and F. Gardoni (2015). "Rabphilin 3A retains NMDA receptors at synaptic sites through interaction with GluN2A/PSD-95 complex." Nat Commun **6**: 10181.

Stanic, J., M. Mellone, F. Napolitano, C. Racca, E. Zianni, D. Minocci, V. Ghiglieri, M. L. Thiolat, Q. Li, A. Longhi, A. De Rosa, B. Picconi, E. Bezard, P. Calabresi, M. Di Luca, A. Usiello and F. Gardoni (2017). "Rabphilin 3A: A novel target for the treatment of levodopa-induced dyskinesias." Neurobiol Dis **108**: 54-64.

Zeng, M., X. Chen, D. Guan, J. Xu, H. Wu, P. Tong and M. Zhang (2018). "Reconstituted Postsynaptic Density as a Molecular Platform for Understanding Synapse Formation and Plasticity." Cell **174**(5): 1172-1187 e1116.

Zhang, X., M. Vigers, J. McCarty, J. N. Rauch, G. H. Fredrickson, M. Z. Wilson, J. E. Shea, S. Han and K. S. Kosik (2020). "The proline-rich domain promotes Tau liquid-liquid phase separation in cells." J Cell Biol **219**(11).

REVIEWERS' COMMENTS

Reviewer #1 (Remarks to the Author):

The present study demonstrates that the GluN2A-specific binding partner Rph3A undergoes phase separation in HEK293 cells, cell-free systems, and neurons. In addition, the study identifies the residues in the N-terminus of Rph3A as essential for its phase separation capability. The original version of the manuscript did not include key experiments in the neuronal system to support the functional role of Rph3A phase separation in driving NMDA receptor synaptic localization and protein-protein interactions. Conversely, the revised version contains several additional experiments performed in primary neuronal cultures that increase the overall value of the present study demonstrating a very important role of Rph3A phase separation also at dendrites and dendritic spine level.

I think that an English proofreading would be useful for improving the overall quality of the text.

Other points:

Even if the authors have extensively revised the Discussion including all new findings, I think that the Discussion section still lacks a relevant issue related to the overall effect of Rph3A on synaptic and extrasynaptic localization of GluN2A-containing receptors. What happens to these receptors if they decrease both at synaptic and extrasynaptic sites? Are they internalized and then degraded? Please discuss this point.

I do not understand the reason why the authors use RPH3A (all uppercase) and not Rph3A as abbreviation of the protein Rabphilin-3A. Please modify throughout the text.

Line 436 – “Our preliminary data showed...” should be modified in “Our data show...”

Reviewer #2 (Remarks to the Author):

The authors have done an impressive job in revising the manuscript both by a new set of experiments and text revisions. The revisions have addressed most of concerns raised by this reviewer. I have two remaining minor suggestions/requests:

1. Line 97: Please change “CTD (aa1244-1464)” to “part of CTD (specifically covering 1244-1464 of the entire C-terminal tail of 839-1464; referred to as GluN2A-CTD from hereon)”
2. For Figures 3,6,7 and Figures S6, S8, S9: the authors showed a segment for each cultured neuron. Please show the corresponding image of the full neuron in the supplemental materials and mark which dendritic segment was selected for detailed analysis. This is an important information for readers to

judge the status of the neurons and the analysis methods used in the experiments.

Reviewer #3 (Remarks to the Author):

The authors have addressed all my concerns and have substantially improved the quality of the submission. I do not have any other concerns and believe the manuscript is ready for publication.

Responses to the reviewers' comments on Yang et al.,
" Rabphilin 3A undergoes phase separation to regulate GluN2A
mobility and surface clustering "

We greatly appreciate the overall positive responses from the reviewers and thank them for pointing out the remaining questions. We have followed the reviewers' suggestions and we hope that the reviewers find the revised paper substantially improved.

Below, the reviewers' comments are in italics, and our responses and the changes made in the revised manuscript are in bold.

Reviewer Comments:

Reviewer #1 (Remarks to the Author):

The present study demonstrates that the GluN2A-specific binding partner Rph3A undergoes phase separation in HEK293 cells, cell-free systems, and neurons. In addition, the study identifies the residues in the N-terminus of Rph3A as essential for its phase separation capability. The original version of the manuscript did not include key experiments in the neuronal system to support the functional role of Rph3A phase separation in driving NMDA receptor synaptic localization and protein-protein interactions. Conversely, the revised version contains several additional experiments performed in primary neuronal cultures that increase the overall value of the present study demonstrating a very important role of Rph3A phase separation also at dendrites and dendritic spine level.

Response:

We are grateful for the reviewer's positive comments.

I think that an English proofreading would be useful for improving the overall quality of the text.

Response:

We thank the reviewer for the suggestion. To improving the overall quality of the text, we have made an English language editing by Scribendi.

Other points:

Even if the authors have extensively revised the Discussion including all new findings, I think that the Discussion section still lacks a relevant issue related to the overall effect of Rph3A on synaptic and extrasynaptic localization of GluN2A-containing receptors. What happens to these receptors if they decrease both at synaptic and extrasynaptic sites? Are they internalized and then degraded? Please discuss this point.

Response:

We thank the reviewer for noticing this weakness. As suggested by the reviewer, we discussed the potential relationship between the phase separation of Rph3A and the endocytosis of GluN2A in line 471 to 480 with the following text: “In this study, we found that disrupting the phase separation of Rph3A impaired the stabilization of synaptic and extrasynaptic GluN2A. Rph3A has been reported to stabilize synaptic GluN2A by blocking endocytosis (Stanic, Carta et al. 2015). These results suggest that disrupting the phase separation of Rph3A might result in the endocytosis of GluN2A in the neuronal surface. The phase separation of certain proteins could form a condense that excludes certain proteins; for instance, the condense formed by the phase separation of excitatory postsynaptic scaffolds could exclude the inhibitory postsynaptic scaffolds Gephyrin (Zeng, Chen et al. 2018). We posit that the phase separation of Rph3A might exclude the proteins that drive the endocytosis of GluN2A, which could then block the endocytosis of GluN2A and stabilize the GluN2A in the neuronal surface.”

I do not understand the reason why the authors use RPH3A (all uppercase) and not Rph3A as abbreviation of the protein Rabphilin-3A. Please modify throughout the text.

Response:

We thank the reviewer for raising this point, and we acknowledge the incorrect usage of RPH3A as abbreviation of Rabphilin-3A. As suggested by the reviewer, we have changed “RPH3A” to “Rph3A” throughout the text and figures.

Line 436 – “Our preliminary data showed...” should be modified in “Our data show...”

Response:

As suggested by the reviewer, we correct the description in line 429 to 431 with the following text: “Our data shows that Rab3a prevents the phase separation of Rph3A and impairs the interaction between GluN2A and Rph3A (Supplementary Fig. 11).”

Reviewer #2 (Remarks to the Author):

The authors have done an impressive job in revising the manuscript both by a new set of experiments and text revisions. The revisions have addressed most of concerns raised by this reviewer. I have two remaining minor suggestions/requests:

Response:

We are grateful for the reviewer's positive comments.

1. Line 97: Please change "CTD (aa1244-1464)" to "part of CTD (specifically covering 1244-1464 of the entire C-terminal tail of 839-1464; referred to as GluN2A-CTD from hereon)"

Response:

We thank the reviewer for pointing out this. As suggested by the reviewer, we corrected the description in line 88 to 90 with the following text: "After phase separation, Rph3A condensed with the part of CTD (specifically covering amino acid (aa) 1244-1464 of the entire C-terminal tail of aa 839-1464; referred to as GluN2A-CTD from hereon) of the GluN2A subunit of the NMDAR."

2. For Figures 3,6,7 and Figures S6, S8, S9: the authors showed a segment for each cultured neuron. Please show the corresponding image of the full neuron in the supplemental materials and mark which dendritic segment was selected for detailed analysis. This is an important information for readers to judge the status of the neurons and the analysis methods used in the experiments.

Response:

As suggested by the reviewer, we showed the corresponding full images of neuronal segments in Fig. 6, 7 and Supplementary Fig. 6, 8 and 9, and marked the selected segments by white rectangles in Supplementary Fig. 13 (Supplementary Fig. 13a-d related to Fig. 6a, 6b, 6c, 6g; Supplementary Fig. 13e-g related to Fig. 7c, 7d, 7h; Supplementary Fig. 13h, 13i related to Supplementary Fig. 6a, 6e; Supplementary Fig. 13j related to Supplementary Fig. 8a; Supplementary Fig. 13k related to Supplementary Fig. 9a). As we used a 100X objective to take the images and we mainly focused on the dendrites of neurons, some of full images may not include the entire neurons (soma and dendrites).

Reviewer #3 (Remarks to the Author):

The authors have addressed all my concerns and have substantially improved the quality of the submission. I do not have any other concerns and believe the manuscript is ready for publication.

Response:

We are grateful for the reviewer's positive comments.

References

Stanic, J., M. Carta, I. Eberini, S. Pelucchi, E. Marcello, A. A. Genazzani, C. Racca, C. Mulle, M. Di Luca and F. Gardoni (2015). "Rabphilin 3A retains NMDA receptors at synaptic sites through interaction with GluN2A/PSD-95 complex." Nat Commun **6**: 10181.

Zeng, M., X. Chen, D. Guan, J. Xu, H. Wu, P. Tong and M. Zhang (2018). "Reconstituted Postsynaptic Density as a Molecular Platform for Understanding Synapse Formation and Plasticity." Cell **174**(5): 1172-1187 e1116.